**Using Spectral Methods to Obtain Particle Size Information from Optical Data:**
**Applications to Measurements from CARES 2010**
Dean B. Atkinson[1], Mikhail Pekour[2], Duli Chand[2], James G. Radney[1,***], Katheryn R. Kolesar[5,*], Qi Zhang[3],
Ari Setyan[3,**], Norman T. O'Neill[4], Christopher D. Cappa[5]
[1] [Department of Chemistry, Portland State University, Portland, OR, USA, 97207]
[2] [Pacific Northwest National Laboratory, Richland, WA, USA, 99352]
[3] [Department of Environmental Toxicology, University of California, Davis, CA, USA, 95616]
[4] [Centre d'Applications et de Recherches en Télédétection, Université de Sherbrooke, Sherbrooke,
Canada]
[5] [Department of Civil and Environmental Engineering, University of California, Davis, CA, USA, 95616]
* Now at: Air Sciences, Inc., Portland, OR, 97214, USA
** Now at: Empa, Swiss Federal Laboratories for Materials Science and Technology, 8600 Dübendorf,
Switzerland
*** Now at: Material Measurement Laboratory, National Institute of Standards and Technology,
Gaithersburg, Maryland, 20899, USA
Correspondence to: D. B. Atkinson (atkinsond@pdx.edu)
## Abstract
Multi-wavelength *in situ* aerosol extinction, absorption and scattering measurements made at two
ground sites during the 2010 Carbonaceous Aerosols and Radiative Effects Study (CARES) are analyzed
using a spectral deconvolution method that allows extraction of particle size-related information,
including the fraction of extinction produced by the fine mode particles and the effective radius of the
fine mode. The spectral deconvolution method is typically applied to analysis of remote sensing
measurements.  Here, its application to *in situ* measurements allows for comparison with more direct
measurement methods and validation of the retrieval approach. Overall, the retrieved fine mode
fraction and effective radius compare well with other *in situ* measurements, including size distribution
measurements and scattering and absorption measurements made separately for $PM_1$ and $PM_{10}$,
although there were some periods during which the different methods yielded different results. One key
contributor to differences between the results obtained is the alternative, spectrally based definition of
"fine" and "coarse" mode from the optical methods, relative to instruments that use a physically
defined cut-point. These results indicate that for campaigns where size, composition, and multi-
wavelength optical property measurements are made, comparison of the results can result in closure or
can identify unusual circumstances. The comparison here also demonstrates that *in situ* multi-
wavelength optical property measurements can be used to determine information about particle size
distributions in situations where direct size distribution measurements are not available.

## Introduction


Aerosols remain a substantial source of uncertainty in climate models, despite considerable progress in
scientific understanding of their chemical, physical and optical properties in the last few decades (IPCC,
2013). As greater understanding has developed in each of these areas, new complexity is also uncovered
and the interconnectedness of the various properties becomes even more evident. Light scattering by
atmospheric particles has a net cooling effect on climate that is one major offset to greenhouse gas
induced climate warming (Charlson et al., 2005; Bond et al., 2011). The efficiency with which the
atmospheric aerosol interacts with electromagnetic radiation (e.g. sunlight) is dependent upon the size,
composition, shape and morphology of the particles. These properties are not static in time, instead
evolving as particles are transported through the atmosphere as a result of chemical processing,
scavenging and changes in the environmental conditions (e.g. relative humidity and temperature)
(Doran et al., 2007; George and Abbatt, 2010; Lack and Cappa, 2010).
Characterization of the spatial distribution of aerosol particle concentrations and properties is important
to assessing their impact on the atmospheric radiation budget through direct aerosol-radiation and
indirect aerosol-cloud interactions. Aerosol optical properties can be measured directly in the laboratory
and in the field using both in situ methods (Andrews et al., 2004; Moosmuller et al., 2009; Coen et al.,
2013) and remote sensing instruments/platforms, such as sunphotometers and satellites (Holben et al.,
1998; Anderson et al., 2005). Alternatively, aerosol optical properties can be inferred from
measurements of particle composition, abundance and size distributions (Atkinson et al., 2015).  One
particular advantage of the remote sensing instruments is that they allow for characterization of
column-average atmospheric particle burdens and properties over a large spatial scale and are free from
sampling biases as the particles are characterized as they exist in the atmosphere. However, they can
only reliably retrieve aerosol properties under cloud-free conditions, and determination of properties
beyond the aerosol optical depth (such as the single scatter albedo or the aerosol size distribution)
typically requires a data 'inversion' process that relies on an assessment of the wavelength-dependent
light attenuation and scattering (Dubovik and King, 2000). *In situ* methods can allow for more detailed
characterization of aerosols, including the relationships between size, composition and optical
properties, but typically at the expense of reduced spatial coverage and with long-term measurements
typically restricted to the surface (Andrews et al., 2004). Given the wide-spread use of aerosol remote
sensing and the extensive availability of the data (in particular from ground-based sunphotometer
networks such as AERONET and AEROCAN (Holben et al., 1998; Bokoye et al., 2001)), continued
assessment and validation of the inversion methods by comparison with measurements by *in situ*
methods is important.
Multi-wavelength optical measurements can yield information about the aerosol size distribution, a
principle that dates back to Ångström's observation that the wavelength-dependence of light
attenuation by particles was weaker for larger particles (diameters of hundreds of nanometers to
micrometers) than for smaller particles (Ångström, 1929). One of the simplest ways of characterizing the
wavelength-dependence of optical measurements (whether extinction, scattering or absorption) is
through the Angstrom exponent. For a pair of optical measurements at different wavelengths, å
= $-\log(b_{x,\lambda 1}/b_{x,\lambda 2})/\log(\lambda_1/\lambda_2)$, where $b_{x,\lambda}$ is the optical coefficient at one of the wavelengths λ; for
scattering and extinction å typically increases as particle size decreases. The dependence of $b_x$ on
wavelength can alternatively be obtained from a $\log(b_{x,\lambda})$ vs. $\log(\lambda)$ plot using two or more wavelengths;
if the dependence is linear, a regression would obtain the same value as the pair-wise treatment, but
non-linearity can be accommodated by using the continuous derivative $\alpha = -d\ln(b_{x,\lambda}) / d\ln(\lambda)$ at a
specified wavelength. A list of the symbols and acronyms used in this work is provided in Appendix A.
The two-wavelength version will be referred to here as the Ångström exponent and the multi-
wavelength variant as the spectral derivative. Particle size classification schemes have been proposed
(Clarke and Kapustin, 2010) and  supported/validated (Eck et al., 2008; Massoli et al., 2009; Cappa et al.,
2016) based on the Ångström exponent of extinction or scattering. When observations are made at
more than two wavelengths (ideally, widely spaced), further information regarding the nature of the
particle size distribution can be extracted. For example, an additional level of refinement of wavelength-
dependent measurements of aerosol optical depth (path integrated extinction) was introduced by
O'Neill et al. (2005) to aid in the interpretation of the data obtained by the ground-based
sunphotometer networks AERONET and AEROCAN. Specifically, O'Neill et al. (2003; 2005) showed that
the fine mode fraction (FMF) of extinction and the fine mode effective radius, $R_{eff,f}$ could be extracted
directly from the multi-wavelength optical depth or extinction measurements available from remote
sensing. The FMF provides for an approximate discrimination between what are typically naturally
produced coarse mode particles (dust or sea spray) and what are often anthropogenically associated
fine mode particles. Thus, parameters such as the FMF can provide a nominal indication of the relative
contributions of natural versus anthropogenic particles to the atmospheric AOD. Variations in $R_{eff,f}$
provide information on the sources of the fine mode particles - as different sources yield fine mode
particles with different size distributions - or the extent to which particles have undergone atmospheric
processing, which can change the size distribution (and chemical composition) in systematic ways.
In the spectral curvature approach of O'Neill et al. (2003), the fine mode spectral derivatives ($\alpha_f$ = first
derivative and $\alpha_f'$ = second) and the FMF are first extracted from multi-wavelength extinction data using
a process described as Spectral Deconvolution. The fine mode spectral derivatives can then be used to
obtain the fine mode effective radius from a fine mode spectral curvature algorithm. Alternatively, the
fine mode effective radius can be calculated from direct measurements of size distribution (e.g. from
scanning mobility particle sizer) using equation 1 (Hansen and Travis, 1974):
$$R_{eff,f} = \frac{\int_0^\infty R\pi R^2 \frac{dN}{dlnR} dlnR}{\int_0^\infty \pi R^2 \frac{dN}{dlnR} dlnR}$$    (1)
where $R$ is the particle geometrical radius and dN/dlnR is a number weighted size distribution for which
$R_{eff,f}$ is the first moment (average radius) of the surface-area weighted size distribution. $R_{eff,f}$ is an
effective radius that characterizes, approximately, the average size of particles in the fine mode that
scatter solar radiation. In this work, we compare the optically obtained $R_{eff,f}$ retrievals to those
calculated by numerically evaluating the integrals of Equation 1 using the observed size distributions
produced by scanning mobility particle sizers. A single log-translatable particle size distribution (i.e., a
PSD that can be translated along the log-transformed particle size axis without changing the form of the
distribution function) is, in many cases, a reasonable representation of the size distribution of observed
aerosol fine modes (O'Neill et al., 2005). In these cases, the fine mode can be characterized by the single
parameter $R_{eff,f}$ facilitating comparisons and examination of trends in sources and/or atmospheric
processing.
Numerical methods such as those developed by O'Neill et al. (2003) were originally applied to remote
sensing measurements, but can also be applied to *in situ* extinction measurements. Beyond adding to
the utility of the *in situ* optical measurements, this provides an opportunity to test the methods against
other, complementary measures of particle size and size-dependent scattering and extinction. For
example, Atkinson et al. (2010) used the approach of O'Neill et al. (2003) to analyze *in situ,* three-
wavelength aerosol extinction measurements made during the 2006 TexAQS II campaign near Houston,
TX. More recently, Kaku et al. (2014) showed, for a range of marine atmospheres, that the application of
this spectral approach to obtain FMF from three-wavelength scattering coefficient measurements was
largely coherent with the sub-micron fraction of scattering (SMF), obtained from scattering coefficient
measurements of the fine and coarse mode components using impactor-based separation of the
aerosol. These studies, and others, provide a useful basis for understanding the accuracy and
applicability of the parameters retrieved from remote sensing data. However, further assessment in a
wide range of environments is necessary given that networks employing such spectral remote sensing
algorithms (AERONET and some surface based sites) represent locations impacted by particles from
diverse sources.
In this work, measurements of aerosol optical properties (extinction, scattering and absorption
coefficients) made at multiple wavelengths during the 2010 Carbonaceous Aerosols and Radiative
Effects Study (Fast et al., 2012; Zaveri et al., 2012) are reported and analyzed using the O'Neill et al.
(2003) and the O'Neill et al. (2008b) methods. The measurements were made at two locations near
Sacramento: a more urban site in Granite Bay, CA (T0) and a more rural site in Cool, CA (T1) that were
often linked by direct atmospheric transport. The multi-wavelength measurements were made using
three types of optical instruments (specifically seven separate instruments at the two locations). The
multi-wavelength measurements of the extinction coefficients (either measured directly or produced
from the sum of scattering and absorption coefficients) are used to retrieve the fine mode fraction of
extinction and fine mode effective radius. These results from the retrieval, described in more detail in
the next section, are compared to other, complementary *in situ* measurements. Scattering and
absorption coefficients were measured after aerodynamic separation into the $PM_1$ and $PM_{10}$ fractions,
which allowed the sub-micron fraction (SMF) of extinction to be directly determined. The *in situ* SMF can
be compared with the FMF from the spectral retrieval method. In this work, sub-micron particles are
those with nominal aerodynamic diameters ($d_{p,a}$) smaller than 1 μm, likely resulting in geometric
diameters below 800 nm. Also, size distribution measurements allowed for determination of the fine-
mode effective radii (via Eqn. 1), which are compared with those obtained from the spectral retrieval.

## Theoretical Approach

### *The Spectral Deconvolution Algorithm with Fine Mode Curvature (SDA-FMC) Approach*

This section provides a qualitative description of the fine and coarse mode AOD (or extinction) retrieval
algorithm (SDA, or spectral deconvolution algorithm) and fine mode optical sizing (FMC or fine mode
curvature) method developed by O'Neill. The details of the derivation and application of the SDA are
provided in previous publications (O'Neill et al., 2003; Atkinson et al., 2010; Kaku et al., 2014). The
MATLAB code that implements the approach is available from O'Neill upon request. Application of both
approaches requires a robust set of measurements of aerosol optical extinction or scattering (or optical
depth) at a minimum of three wavelengths that should be widely spread across the optical region of the
spectrum (near UV through the visible to the near IR; see, for example, O'Neill et al. (2008a)).
The fundamental assumption of the SDA approach is that most ambient aerosol size distributions are
composed of two optically-relevant modes: a fine mode having an effective radius (and to a lesser
extent, geometric standard deviation) that is a function of atmospheric processing, and a separate
coarse mode, largely in the supermicron ($d_{p,a}$ > 1 µm) size range. A common assumption is that the fine
mode is more closely associated with anthropogenic activities and the coarse mode with natural
sources, although this can be somewhat confounded by smoke from biomass burning (Hamill et al.,
2016). In particular, it can be difficult to distinguish biomass burning particles from particles derived
from urban sources, as both primarily fall within the fine mode and are somewhat absorbing. The FMC
(Fine Mode Curvature) algorithm employs the fine mode optical parameters retrieved using the SDA to
estimate both a fundamental indicator of optical particle size (the fine mode van de Hulst parameter)
and from this, an indicator of microphysical particle size (the fine mode effective radius); these are both
defined below.
***Spectral deconvolution of the fine and coarse mode extinction and derivation of the fine mode***
***spectral derivatives (SDA)***
The spectral deconvolution algorithm begins by isolating the fraction of total extinction due to particles
in the fine mode, based on the stronger dependence of the extinction (scattering)[1] on wavelength for
smaller particles. Current applications of the method start by fitting ln($b_{ext}$) (or ln($b_{scat}$) or ln(AOD))
versus ln($\lambda$) to a second order polynomial, where $b_{ext}$ is the measured wavelength-dependent extinction
coefficient (see Atkinson et al. (2010) and Kaku et al. (2014) for scattering and extinction coefficient
applications, Saha et al. (2010) for a sunphotometry AOD application and Baibakov et al. (2015) for a
starphotometry AOD application). The extinction and its first and second derivatives are determined
from the fit at a reference wavelength of 500 nm, a common reference wavelength along with 550 nm
in optical studies. The first derivative (i.e. slope) is denoted $\alpha$ in analogy to the Ångström exponent, but
in this non-linear, second order approach it is a function of wavelength. The second derivative $\alpha'$ (i.e.
spectral curvature) may, in principle, be wavelength dependent over the observed range, but using a
second order polynomial fit yields a constant value. Values of $\alpha$ and $\alpha'$ associated with the fine mode

---

[1] We will stop inserting "(scattering)" at this point although all references below should be understood to apply to both scattering and extinction.

and the coarse mode are indicated using subscript f or c, respectively. In this work, only a second order
fit is possible because only three measurements are used to define the wavelength dependence. In the
SDA-FMC approach, the observed spectral derivative ($\alpha$) is used along with the SDA-derived fine mode
spectral derivative ($\alpha_f$) to produce the fine mode fraction of extinction (FMF), given as:

$$FMF = \frac{\alpha - \alpha_c}{\alpha_f + \alpha_c}$$  (2)

Ultimately, the fine mode slope and curvature are both used in the FMC algorithm to determine the fine
mode effective radius (discussed in the next section).
The algorithm proscribes constant values of the spectral slope and curvature for all coarse mode
aerosols ($\alpha_C$ and $\alpha'_C$) at the reference wavelength of 500 nm. Specifically, $\alpha_C = -0.15 \pm 0.15$  and $\alpha'_C =$
$0.0 \pm 0.15$, with the uncertainties as per O'Neill et al. (2003). O'Neill et al. (2001) showed that an
assumption of an aerosol size distribution with two distinct modes yields a series of three equations that
express the relationships between the observed parameters (AOD or extinction coefficient, $\alpha$, $\alpha'$) and
their fine and coarse mode analogues. Specifically, the equations can be inverted to yield the fine mode
spectral derivative, the fine mode curvature ($\alpha_f'$) and the fine and coarse mode AOD or $b_{ext}$ values. It
should be noted that the fitting of a $2^{nd}$ order polynomial to input AOD or $b_{ext}$ spectra is only an
approximation relative to a higher order polynomial. The use of a $2^{nd}$ order polynomial represents a
compromise between higher order spectral polynomials being better representations of theoretical Mie
spectra and the beneficial damping effects of lower order polynomials in the presence of noisy spectra
(O'Neill et al., 2001). The observationally determined total and fine mode spectral derivative and
proscribed coarse mode spectral derivative are then used to calculate the fine mode fraction of
extinction at the reference wavelength (here 500 nm) using Eqn. 2.

*Estimation of the Fine Mode Effective Radius – the Fine Mode Curvature (FMC) approach*
Using the SDA-derived, fine mode spectral derivatives ($\alpha_f'$ and $\alpha_f$), an estimate of the fine mode effective
radius is obtained. The basis for this approach is a fundamental parameterization involving the effective
van de Hulst phase shift parameter for fine mode aerosols and its representation in $\alpha_f'$ versus $\alpha_f$ space.
Full details are provided in O'Neill et al. (2005) and O'Neill et al. (2008b), and only a summary of the
parameterization is provided here. The van de Hulst parameter for the fine mode, *peff,f*, is given by:
$\rho_{eff,f} = 2 * \frac{2\,\pi R_{eff,f}}{\lambda} |m - 1|$                (3)

where λ is the reference wavelength and m is the complex refractive index at that wavelength (O'Neill et
al., 2005). An estimate of this purely optical parameter is based on a 3$^{rd}$ order polynomial derived from
numerical Mie simulations that relate $\rho_{eff,f}$ and the polar angle (ψ) coordinate of any point in $\alpha_f'$ vs. $\alpha_f$
space (O'Neill et al., 2005). The value of ψ for any given retrieval is simply the arctangent of $\alpha_f'$ divided
by $\alpha_f$ (minus small prescribed offsets of $\alpha_{f,0}'$ over $\alpha_{f,0}$ respectively).  Individual simulated contour curves
of $\alpha_f'$ versus $\alpha_f$ correspond to particle size distributions of differing $R_{eff,f}$ for constant values of refractive
index and were illustrated in Figure 1 of O'Neill et al. (2005). The three different "lines of constant $\rho_{eff,f}$"
in that figure correspond to three different values of ψ (where both $\rho_{eff,f}$ and ψ increase in the
counterclockwise direction from the horizontal). The $R_{eff,f}$ value are then computed from the retrieved
value of $\rho_{eff,f}$, by inverting equation (3), if the refractive index of the particles is known.  Since the
refractive index is generally unknown for the situations we consider here, the information provided by
this approach is actually a combination of size and composition. In many cases, an average, constant
value for the real portion of the refractive index can be assumed and the imaginary part neglected to
provide an estimate of the effective radius; this is, in part, because the imaginary component is typically
much smaller than the real component of the refractive index, and thus the $R_{eff,f}$ value is relatively
insensitive to variations in the imaginary component. This treatment is questionable if strong changes in
the average composition that lead to changes in m are suspected. For example if the composition
shifted from pure sulfate aerosol (*m* = 1.53 + 0i) to a brown carbon organic (*m* = 1.4 + 0.03i) this would
introduce a 33% shift in the derived radius with no change in actual size; the majority of this shift in the
derived radius results from the change in the real component of the refractive index.
The FMC method represented by the inversion of equation (3) has been less rigorously validated than
the SDA portion and is expected to be more susceptible to problems related to measurement errors and
a decreasing sensitivity with decreasing fine mode fraction of extinction. The FMC validation is largely
confined to comparisons with the more comprehensive AERONET inversions of Dubovik and King (2000),
referred to henceforth as the D&K inversions. These inversions, which require the combination of AOD
and sky radiance data, are of a significantly lower frequency than simple AOD measurements. The sky
radiance data are collected nominally once per hour while AOD measurements are made once every 3
minutes. Comparisons between the FMC method and the D&K inversions show averaged FMC versus
AERONET differences of 10% $\pm$ 30% (mean $\pm$ standard deviation of ($\rho_{eff,f,FMC}$ - $\rho_{eff,f,D\&K}$) / $\rho_{eff,f,D\&K}$) for large
FMF values > 0.5, at least for the limited data set of O'Neill et al. (2005) and confirmed by more recently
unpublished AERONET-wide comparisons between the FMC and D&K methods.

## Application of the SDA-FMC method to in situ extinction measurements

This paper seeks to address the following two key questions pertaining to the use of the SDA-FMC
algorithm with extinction measurements, especially those produced by the cavity ring-down
instruments, to extract information about aerosol size, both the partitioning of the extinction between
the fine and coarse modes and the extraction of a single parameter size characterization of the fine
mode.
1.) Can the approach be used reliably to extract the fine and coarse mode fractions of the

256       extinction in situations where only a single optical instrument is used?

and,
2.) In situations where complementary measurements (mobility-based sizers, parallel or switching

259       nephelometers, etc.) are available, what information can be determined from the comparison of

260       the products of the SDA-FMC approach to comparable information obtained in other ways?

It has been suggested  that a single multi-wavelength optical measurement of the fine mode fraction
could be less expensive than derivation of the sub-micron fraction of scattering using parallel
nephelometers (Kaku et al., 2014). The use of two size-selected inlets (e.g., 1 and 10 µm cyclones) and
parallel nephelometers is not prohibitively expensive, but the typical concerns regarding calibration
maintenance and careful and consistent application of correction factors for truncation angle and non-
Lambertian illumination can be magnified when measurements are combined (either as differences or
ratios) since systematic errors may not undergo partial cancellation like random errors.
In principle, the use of two parallel CRD extinction measurements could mitigate some of the possible
errors with parallel nephelometers. Cavity ring-down measurements directly quantify total extinction
within the cavity, which is contributed from both gases and particles (Smith and Atkinson, 2001; Brown,
2003). To determine extinction by aerosols only, the entering air stream is periodically directed through
a filter such that a gas-only reference is determined. Extinction by aerosol particles is determined
relative to this gas zero. The aerosol extinction is further corrected to account for the practical aspect
that the complete mirror-to-mirror distance of the optical cavity is typically not filled with aerosols (to
keep the mirrors clean) (Langridge et al., 2011). The former (zeroing) limits instrument precision and
sometimes accuracy while the latter (path length) limits instrument accuracy. In general these
procedures are identical for the two parallel instruments and are very stable in time, so they would only
be expected to produce a small and consistent bias. To our knowledge, currently no single-package,
multi-wavelength direct extinction (cavity-enhanced) instruments are commercially available. Multiple
single-wavelength instruments operating at different wavelengths could be deployed, but might be
prohibitively expensive.
For detailed knowledge of the fine mode size distribution, the use of scanning mobility analyzer-based
sizing instruments is preferable since the full mobility size distribution is obtained, as opposed to only
the effective radius provided by the FMC procedure. However, scanning mobility sizer instruments
typically have maximum diameters of only 700 to 800 nm, and both scanning and multi-channel variants
are of comparable expense and complexity as CRD instruments. In order to obtain additional
information about the coarse mode size distribution and contribution to the optical effects, an aerosol
particle spectrometer (APS) is generally added to the measurement suite.
The purely spectrally-based mode separation inherent in the SDA obviates the need for a physical cut
point selection, such as that required to measure the $PM_1$ scattering product used in this work. This can
be advantageous, since selection and maintenance of a size cut-point is a possible source of differences
between some measurements (and variability of all measurements using physical separation) of the sub-
micron fraction (SMF) of scattering, absorption or extinction. The SMF is fundamentally different from
the FMF, although both provide an indication of the fractional optical contribution of smaller particles.
In fact, there are fundamental differences between many of the SMF or FMF data products that are
currently available. For example, the Dubovik and King (2000) SMF data product tries to locate the
separation radius (called the inflection point) at a minimum of the particle size distribution obtained
from the inversion procedure. This results in a variable cut point that can be interpreted as assigning a
portion of the coarse mode to the fine mode (O'Neill et al., 2003). The aerodynamic diameter selected
for the physical separation used in the SMF presented in this work might result in some mis-assignment
of fine mode extinction to the coarse mode, since (i) the aerodynamic separation results in a cut point
that is less than 1 μm geometric diameter and (ii) the cut point might not correspond to a local
minimum of the size distribution. These definitional differences should be kept in mind when comparing
fine mode apportionments (SMF or FMF) from different measurements/data treatments. And all of
these data products will usually differ significantly from the optical properties of the $PM_{2.5}$ fraction used
to define the fine mode for air quality regulations and to exclude larger particles in the CRD instruments
at T0. The latter allowed a significant fraction, but not all of the optically coarse particles into the
instruments, as shown in the Results section. For the comparisons presented in this work, in cases
where there is significant penetration of one of the modes into the size regime defined by the physical
cut-point as the other mode (or significant overlap of two or more size modes) there are noticeable
differences between the physically-defined SMF and the FMF produced by the SDA.

## Experimental

The instrument suites used, sampling conditions and methodology and goals of the CARES study have
been summarized by Zaveri et al. (2012). A summary of the instrumentation used to make the light
extinction, scattering and absorption measurements is provided in Table 1. Extinction was measured
either directly (using cavity ringdown spectroscopy) or as the sum of scattering and absorption. A brief
description of the key instruments used in the current analyses is given below.

**Table 1: Summary of optical instruments used at the T0 and T1 sites**

| Property | Instrument | Wavelength | Size Cut[*] |
|---|---|---|---|
| | | *T0* | |
| Extinction | UCD CRD | 405, 532 nm | 2.5 μm |
| | PSU CRD | 532, 1064 nm | 2.5 μm |
| Scattering | PNNL Nephelometer | 450, 550, 700 nm | 1 μm, 10 μm |
| Absorption | PNNL PSAP | 470, 522, 660 nm | 1 μm, 10 μm |
| | | *T1* | |
| Extinction | PSU CRD | 355, 532, 1064 nm | None applied |
| Scattering | PNNL Nephelometer | 450, 550, 700 nm | 1 μm, 10 μm |
| Absorption | PNNL PSAP | 470, 522, 660 nm | 1 μm, 10 μm |

*For the entries with two size cuts listed, the sampling system switched
between the two on a 6 minute cycle


***Instruments used at the T0 site (American River College, Granite Bay, CA USA)***
Cavity Ring-down Extinction: The $b_{ext}$ measurements at 405 nm and 532 nm were made using the UC
Davis two-wavelength Cavity Ring Down-Photoacoustic Spectrometer (CRD-PAS) instrument (Langridge
et al., 2011; Lack et al., 2012). Full details of these measurements are available in Cappa et al. (2016) and
Atkinson et al. (2015). These measurements were only made for a subset of the CARES campaign, from
20:00 PDT on 16 June through 09:00 PDT on 29 June. At 532 nm, $b_{ext}$ was measured at low (~25%), mid
(~75%) and high (~85%) relatively humidity. At 405 nm only low RH measurements were made, and so
only the low RH 532 nm measurements are used in this study. The CRD-PAS sampled behind a $PM_{2.5}$
(aerodynamic diameter <2.5 μm) URG Teflon-coated aluminum cyclone. A separate CRD instrument
deployed by the PSU group at T0 used a single optical cavity to measure the sub-2.5 μm (sampled
through a similar URG cyclone) aerosol extinction coefficient at 532 and 1064 nm simultaneously
(Radney et al., 2009). This instrument did not incorporate intentional RH control, but efforts were made
to maintain nearly ambient conditions, resulting in low RH (25 - 40 %) throughout most of the campaign,
as measured by an integrated RH/T sensor (Vaisala HMP70). Daytime ambient RH was similar to the low
RH value during the CARES campaign (Fast et al., 2012).
To obtain three-wavelength $b_{ext}$ measurements for use in the SDA-FMC analysis, we combined the
measurements from the two CRD instruments (the 1064 nm measurements from the PSU instrument
were used with the 532 nm and 405 nm UCD data after all had been averaged to one-hour). To assess
whether this was a reasonable approach, the 532 nm time series data from the two instruments were
overlaid and examined for differences. There is a high degree of temporal correspondence between the
measurements from the two instruments, although there was a clear difference in precision, with the
UCD CRD having approximately 3 times better precision than the PSU instrument at comparable
integration times. This difference in precision results from differences in instrumental design and (likely)
mirror quality. A scatterplot (Figure S1) of $b_{ext,PSU}$ versus $b_{ext,UCD}$ also showed good correlation, with a best
fit line from an orthogonal distance regression fit having a slope = 0.96 and an intercept that was
statistically indistinguishable from zero. This is within the uncertainties of the instruments. The good
agreement at 532 nm between the PSU and UCD instruments suggests that combining the 1064 nm
measurements from PSU with the 405 nm and 532 nm measurements from UCD is reasonable. If the
very slight low bias in the 532 nm $b_{ext}$ from PSU relative to the UCD measurements applies to the 1064
nm measurements then the derived FMF values might be slightly overestimated.

Size-selected absorption and scattering (nephelometer and PSAP): The low RH scattering and absorption
coefficients were alternatingly measured for $PM_{10}$ and $PM_1$ aerodynamic size selected aerosol using the
PNNL Aerosol Monitoring System, a clone of NOAA/CMDL's Aerosol Monitoring System (detailed
description at http://www.esrl.noaa.gov/gmd/aero/instrumentation/instrum.html and in Zaveri et al.
(2012)). The relevant measurements are: light absorption coefficients at three-wavelengths (Radiance
Research Particle Soot Absorption Photometer [PSAP]) and total scattering coefficients (three-
wavelength nephelometer, TSI 3563). The scattering coefficients were corrected for truncation error
(Anderson and Ogren, 1998) and the absorption coefficients for filter effects (Ogren, 2010). The
absorption coefficients were interpolated to the nephelometer wavelengths assuming the inverse
wavelength dependence characteristic of uncoated black carbon, as appropriate for this region (Cappa
et al., 2016). The absorption and scattering coefficients for $PM_1$ or $PM_{10}$ are then summed after
averaging to one-hour intervals and using the mean of the 450 and 550 nm values to obtain $b_{ext}$(500
nm). The extinction fraction of the $PM_1$ (herein, the SMF) at the visible wavelength (500 nm) is then
calculated from their ratio
$$SMF_{ext} = \frac{b_{ext,PM1}}{b_{ext,PM10}}$$                      (4)
Particle size control was effected by 2 impactors (1 µm and 10 µm) upstream of the PSAP and
nephelometer. The 10- µm impactor was always present in the sampling line, and the flow was switched
to run through the 1- µm impactor on 6-min intervals, yielding alternating 6-min measurements of
submicron and coarse (< 10 µm) particle modes.
Fine particle size distribution: The submicron dry particle mobility diameter ($d_{p,m}$) size distribution (12
nm to 737 nm) was measured using a scanning mobility particle sizer (SMPS) comprised of a charge
neutralizer, differential mobility analyzer and condensation particle counter (TSI 3081 DMA column and
model 3775 CPC). The SMPS data were corrected for multiply-charged particles and diffusional losses.
These size distribution measurements are used to calculate $R_{eff,f}$ values from Eqn. 1, which will be
referred to as $R_{eff,f,size}$. It should be noted that a mobility diameter of 737 nm corresponds to an
aerodynamic diameter of 919 nm (assuming a density of 1.5 g cm$^{-3}$, a reasonable value for the campaign
based on the observed particle composition (Atkinson et al., 2015)).

***Instruments used at the T1 site (Evergreen School, Cool, CA USA)***
Cavity Ring-down Extinction: The PSU group deployed a custom CRD instrument that used separate
optical cavities to measure $b_{ext}$ at 355 nm, 532 nm, and 1064 nm simultaneously in each of four separate
flow systems that were intended to measure total and submicron aerosol and submicron aerosol that
had been conditioned to have elevated and suppressed RH. Only the total aerosol flow results are used
here as this prototype system suffered from signal to noise problems and RH/temperature control
issues. As with the T0 PSU instrument, the total aerosol system attempts to measure particle extinction
at nearly ambient conditions, resulting in low RH (25 – 40 %) throughout most of the campaign, as
measured by an integrated RH/T sensor (Vaisala HMP70). No intentional size cut was applied to these
measurements, although the system was not optimized for transmission of coarse mode particles.
Size-selected absorption and scattering (Nephelometer and PSAP): An identical instrument suite to that
used at T0 was deployed and the same data analysis was conducted.
Fine particle size distribution: The SMPS used at T1 is a similar design described in (Setyan et al., 2012)
and it measured low RH particle sizes from 10 nm to 858 nm. The SMPS data were corrected to take into
account the DMA transfer function, the bipolar charge distribution, the CPC efficiency and the internal
diffusion losses (Setyan et al., 2014).
***Uncertainties in the derived and measured values***
The uncertainty in the SMF has been estimated from standard error propagation of the uncertainties in
the $PM_1$ and $PM_{10}$ extinction measurements. The assumed uncertainties in $b_{ext,PM1}$ and $b_{ext,PM10}$ are ±1
$Mm^{-1}$. This uncertainty estimate accounts only for random errors, not systematic errors.
Uncertainties in the FMF have been estimated based on the uncertainties in the inputs to the SDA-FMC
procedure, namely the $b_{ext}$ values. The assumed uncertainties in the input $b_{ext}$ were instrument specific:
<1 $Mm^{-1}$ for the UCD CRD, 1 $Mm^{-1}$ for the nephelometer plus PSAP and PSU CRD at T0, and 3 $Mm^{-1}$ for
the PSU CRD at T1. The input uncertainties are propagated through the various mathematical
relationships using standard methods. The FMF error estimate includes some of the factors that
contribute systematic uncertainty in the method. As noted in the Theoretical Approach section, FMF
values from the SDA-FMC procedure have been shown to agree well with those determined from the
more comprehensive inversion method of Dubovik and King (2000).
Uncertainties in the derived $R_{eff,f}$ are also estimated from the uncertainties in the input values. The size-
distribution derived $R_{eff,f}$ values depend on the SMPS measurements. The SMPS instruments were
calibrated (using 200 nm polystyrene latex spheres) prior to the campaign and a drier was used to keep
the aerosol RH < 30% throughout the entire campaign. Periodic checks throughout the campaign
indicate consistent sizing performance to within 5%. The size distribution data used here were corrected
for DMA transfer function, the bipolar charge distribution, the CPC efficiency and internal diffusion
losses. Under these conditions the estimated uncertainties for $D_p$ are around 10% for the size range
between 20 and 200 nm (Wiedensohler et al., 2012). Although larger uncertainties could exist for
smaller and larger particle sizes, the derived $R_{eff,f}$ values fell primarily in this range. The estimated SMPS
uncertainty (Wiedensohler et al., 2012) was estimated based on intercomparisons between different
SMPS instruments and thus probably represents both determinate and indeterminate errors. The
relative uncertainty in the $R_{eff,f}$ from the size distribution measurement is thus estimated to be 10%.
This estimate mainly reflects uncertainties in the absolute size, since there is expected to be significant
cancellation in the errors produced by the particle counter (the same data are used in the numerator
and denominator of Eq. 1).
Estimating the uncertainty in the $R_{eff,f}$ from the SDA-FMC is more challenging because the uncertainties
cannot be simply propagated through the equations. Therefore, an approach was taken wherein a large
number of $R_{eff,f}$ values were calculated from input $b_{ext}$ that were independently, randomly varied within
one standard deviation of the measured value, assuming a normal distribution of errors. Potential
uncertainty or variability in the real refractive index was accounted for based on the compositional
variation (Atkinson et al., 2015) and assuming volume mixing applies. The standard deviation (1s) was
0.015. This is likely a lower estimate of the uncertainty in the RI, as it does not account for absolute
uncertainty in the estimate. The standard deviation of the derived $R_{eff,f}$ is taken as the uncertainty. This
Monte Carlo-style approach does not incorporate systematic error sources. The relative uncertainty in
the derived $R_{eff,f}$ is found to range from a few percent up to 40%, depending on the particular instrument
suite considered and measurement period. In general, the uncertainties were larger for the PSAP and
nephelometer, presumably because the wavelengths used are more closely spaced.

# Results and Discussion

*Fine mode fraction of extinction*
The CRD-based extinction measurements were used to derive the $FMF_{ext}$ using the SDA. This will be
referred to as the $FMF_{ext,CRD}$. For the T0 site, the $FMF_{ext,CRD}$ is for $PM_{2.5}$ while at T1 no physical cut point
was introduced, so $PM_{10}$ is a reasonable expectation. The time series of the CRD-based $b_{ext}$ values and of
the derived $FMF_{ext,CRD}$ at the T0 site are shown in Figure 1 (all times in PDT – local time during the study).
The $FMF_{ext,CRD}$ varies from 0.54 to 0.97, with a mean of 0.79 ± 0.1 (1 σ) as summarized in Table 2.


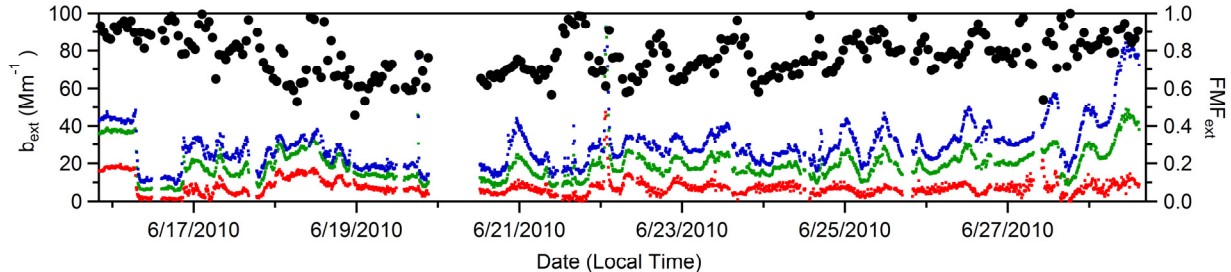


**Figure 1** – Time series of CRD extinction coefficient observations (left axis) and the derived $FMF_{ext,CRD}$ (right axis) at T0 during the time period analyzed in this work. The blue, green and red traces are the 405 nm, 532 nm and 1064 nm $b_{ext}$ (respectively) and the black points show the 1 h average $FMF_{ext,CRD}$ from the SDA analysis. A $PM_{2.5}$ size cut was applied during the sampling.


The fine mode fraction of extinction at T0 was alternatively determined from the $PM_{10}$ $b_{ext}$
measurements from the nephelometer and PSAP, referred to as $FMF_{ext,sum}$. The SDA-derived $FMF_{ext,CRD}$
and $FMF_{ext,sum}$ values are compared with the sub-micron fraction of extinction determined from the
combined $PM_1$ and $PM_{10}$ nephelometer and PSAP measurements (from the latter part of the campaign)
at T0 (Fig. 2). The $FMF_{ext,CRD}$, $FMF_{ext,sum}$ and $SMF_{ext,sum}$ all exhibit the same general temporal dependence.
In general, the $FMF_{ext,CRD} > FMF_{ext,sum} \sim SMF_{ext,sum}$ although the specific relationships vary with time. For
example, there are periods when the $FMF_{ext,sum}$ and $SMF_{ext,sum}$ are nearly identical (e.g. 20 June – 23
June) and periods when the $SMF_{ext,sum}$ is somewhat lower than the $FMF_{ext,sum}$ (e.g. 24 June – 25 June).

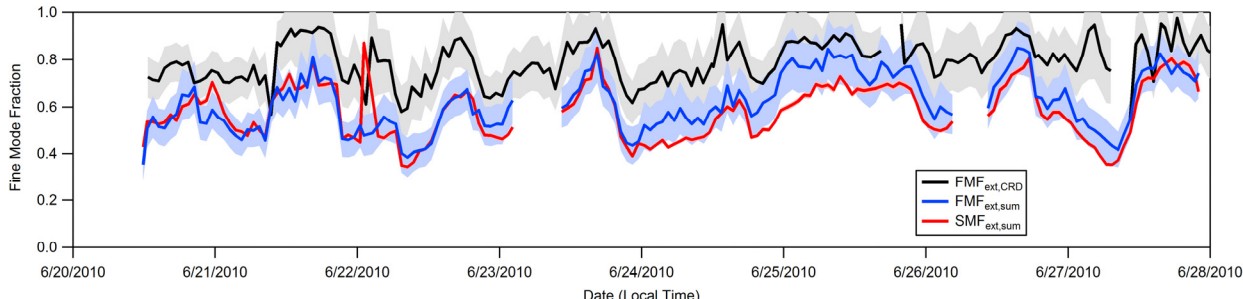


**Figure 2** – Time series of the fine mode fractions and sub-micron fraction of extinction at T0. The
red trace is the $SMF_{ext,sum}$ determined from the $b_{ext}(PM_1) / b_{ext}(PM_{10})$ ratio. The black and blue
traces are the $FMF_{ext}$ from the SDA analysis of the CRD extinction (black) and nephelometer +
PSAP extinction (blue). The $FMF_{ext,CRD}$ values are the same as those of Fig. 1 for the latter half of
the campaign. Uncertainty ranges are shown as light colored bands. The uncertainty of SMF is
only slightly wider than the heavy line that was chosen to represent it.

The $FMF_{ext,CRD}$ was determined for $PM_{2.5}$ while the $FMF_{ext,sum}$ was determined for $PM_{10}$. If a substantial
fraction of the scattering was contributed by particles with diameters >2.5 μm, then the $FMF_{ext,CRD}$
should be larger than the $FMF_{ext,sum}$, as was observed. Kassianov et al. (2012) used measured particle size
distributions from CARES to show that supermicron particles contributed significantly to the total
scattering, consistent with the observation that $FMF_{ext,CRD} > FMF_{ext,sum}$. Variability in the difference
between the $FMF_{ext,CRD}$ and $FMF_{ext,sum}$ likely reflects variability in the contribution of these larger particles
to the total scattering.
The $FMF_{ext,CRD}$, $FMF_{ext,sum}$ and $SMF_{ext,sum}$ were similarly determined from the measurements at the T1 site
(Figure 3). For T1, the CRD measurements were made for particles without any intentional size cut
applied, as opposed to the $PM_{2.5}$ size cut used for the T0 measurements. At this downwind site the
$SMF_{ext,sum}$ , $FMF_{ext,CRD}$ and $FMF_{ext,sum}$ were all very similar, both in terms of the absolute magnitude and
the temporal variability. The $FMF_{ext,CRD}$ ranged from 0.22 to 0.89, with a mean of 0.58 ± 0.16. That the
$FMF_{ext,CRD}$ and $FMF_{ext,sum}$ are very similar in absolute magnitude for T1 but differ at T0 (while still
exhibiting similar temporal variability) is likely related to the application of an intentional size cut for the
CRD measurements at T0 but not at T1. The observations suggest that the T1 CRD without the size cut
samples coarse-mode particles with a similar efficiency as the nephelometer and PSAP having the $PM_{10}$
size cut.
Overall, these results indicate that the use of the spectral deconvolution algorithm on optical data can
robustly provide information on the fine mode fraction of extinction. Moreover, since the $FMF_{ext}$ results
at T1 are similar for the two types of extinction measurements, it seems that the narrower wavelength
range of the nephelometer (450, 550, 700 nm) and PSAP (470, 522, 660 nm) compared to the CRD
instruments used here is still adequate to define the spectral dependence of extinction for extraction of
the slope and curvature parameters. However, the results demonstrate that the optical method does
not allow for a precise definition of "fine" and "coarse" in terms of a specific, effective size cut that
distinguishes between the two regimes. While the SMF has an explicitly defined size cut ($PM_1$), the
effective size cut for the FMF can vary. The effective size cut is dependent on the shapes (i.e. widths,
positions and number of actual modes) of the size distributions in the "fine" and "coarse" size regimes
and the extent of overlap between them, which is dependent on the size range of particles sampled (e.g.
$PM_{2.5}$ versus $PM_{10}$). For remote sensing measurements, the particular size that distinguishes between
the fine and coarse mode therefore likely varies between locations and seasons. Nonetheless, since the
major sources of fine and coarse mode particles are likely to be reasonably distinct in many
environments, the $FMF_{ext,CRD}$ provides a reasonable characterization of the variability in the
contributions of such sources to the total extinction and, in environments where the extinction is
dominated by scattering (i.e. when the SSA is large), to the total scattering as well.

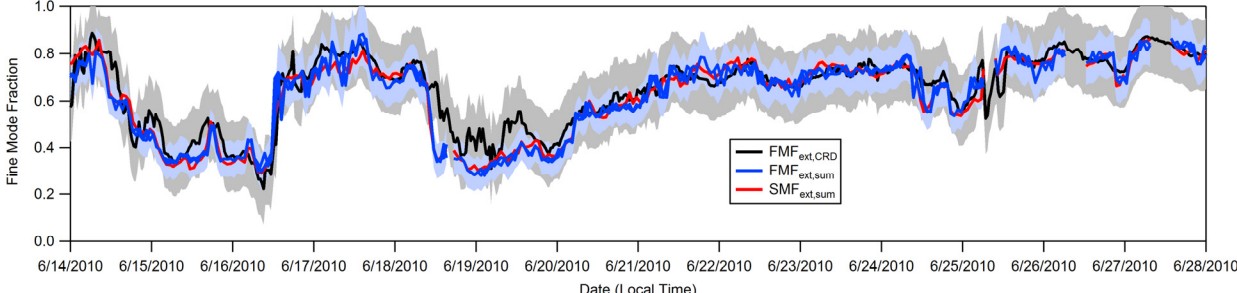


**Figure 3** – the fine mode fraction of extinction (SMF and $FMF_{ext}$) for the latter half of the
campaign at T1. Here, the $FMF_{ext,CRD}$ is determined for particles sampled without a size cut
applied. Uncertainty ranges are shown as light colored bands.

*Effective fine mode radius product of SDA-FMC*
The SDA-FMC analysis also allows for derivation of the fine mode effective radius, $R_{eff,f}$, via Eq. 3.
Determination of $R_{eff,f}$ requires knowledge of the real and imaginary parts of the refractive index. Here,
an average value of $m_r$ = 1.5 is used, based on Atkinson et al. (2015), and absorption is assumed to be
negligible. The latter is a reasonable assumption given the relatively high single scatter albedo values at
the two sites (Cappa et al., 2016), and because assuming the particles to be slightly absorbing has
minimal influence on the results. Temporal variability in $m_r$ due to variability in particle composition will
contribute to uncertainty in the retrieved $R_{eff,f}$. As discussed above, a change in $m_r$ of 0.13 corresponds
approximately to a shift in $R_{eff,f}$ by 30%. The actual variability in $m_r$ is not known for the particles here,
but we expect a shift of 0.13 in $m_r$ to be a reasonable upper limit on physical grounds.
Values of $R_{eff,f}$ are determined using both the CRD-measured $b_{ext}$ and the $PM_{10}$ $b_{ext}$ from the
nephelometer + PSAP measurements for both T0 and T1 (Figure 4). $R_{eff,f}$ values are also determined from
the $PM_1$ nephelometer + PSAP measurements at both sites. Comparison of the $R_{eff,f}$ values between the
$PM_{10}$ and $PM_1$ measurements provides a test of the robustness of the overall retrieval method. The $R_{eff,f}$
from the CRD measurements will be referred to as $R_{eff,f,CRD}$ and from the nephelometer + PSAP as $R_{eff,f,sum}$.
Comparator values of $R_{eff,f}$ were also calculated from the observed mobility size distributions using Eqn.
1, and are referred to as $R_{eff,f,size}$.

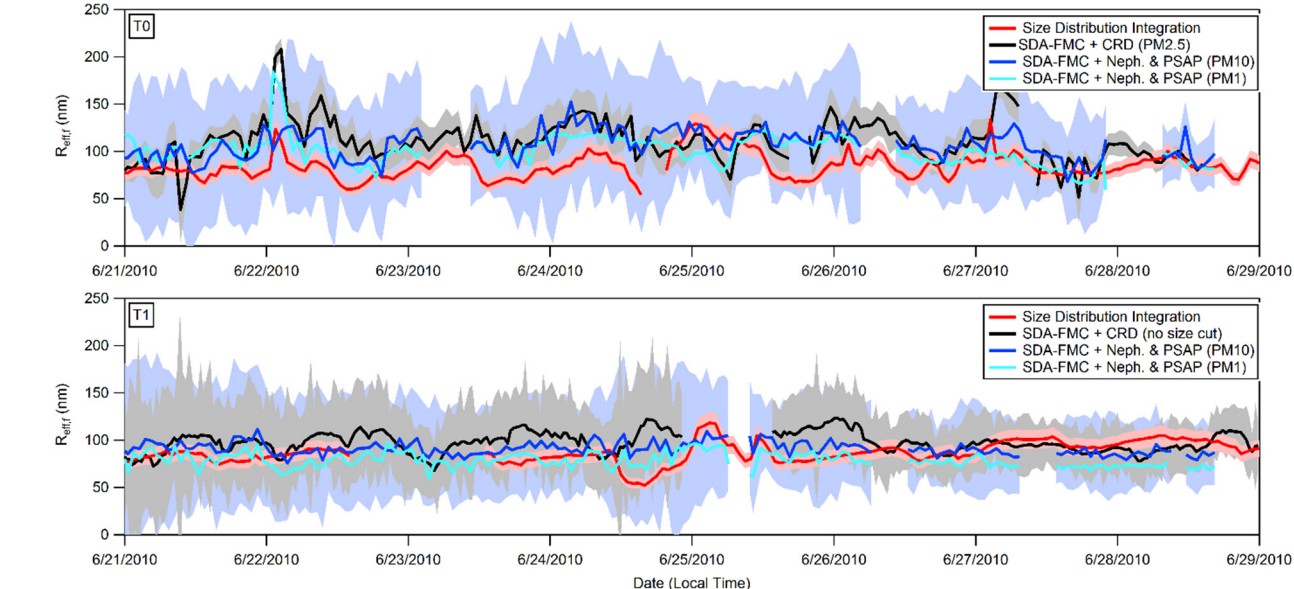



**Figure 4** – Time series of the effective fine mode radii, $R_{eff,f}$, produced by the SDA-FMC analysis of the
CRD data (black) and the nephelometer + PSAP data (blue) from T0 (top) and T1 (bottom). For the
nephelometer + PSAP observations, separate results are shown using either the $PM_{10}$ (dark blue) or
$PM_1$ (light blue) observations. The $R_{eff,f}$ values determined from the size distribution measurements (i.e.
from Eqn. 1) are shown in red.  Uncertainty ranges are shown as light colored bands for each method;
for the SDA-FMC the uncertainty range is only shown for $PM_{10}$ to avoid clutter, but the uncertainty
range is similar for $PM_1$.

The SDA-FMC-derived $R_{eff,f}$ values from the CRD and from the nephelometer + PSAP exhibit reasonably

good agreement in terms of the absolute values and the temporal variability at both the T0 and T1 sites

(Table 2, Fig. 4). Notably, there is good agreement between the $R_{eff,f,sum}$ values obtained from the $PM_{10}$

and $PM_1$ measurements. This provides an important validation of the SDA-FMC procedure, since the

coarse mode contribution to the $PM_{10}$ extinction is substantial and highly variable (Figure 2 and Figure

3).

At T0, the derived $R_{eff,f}$ values range from approximately 70 nm to 140 nm (Table 2), with a few short-

duration periods when $R_{eff,f}$ is outside this range, reflecting short-duration variability in the particle

sources. At T1 the derived $R_{eff,f}$ are generally less variable, ranging from approximately 65 nm to 110 nm,

with fewer particularly low or high periods. The mean $R_{eff,f}$ values between the two sites are similar

(Table 2). At T0, there is a fair degree of temporal coherence of the SDA-FMC results and those obtained

from integration of the size distributions. The generally good temporal agreement between the

optically- and size-derived $R_{eff,f}$ values are even observed during periods where the changes in radius

happened rapidly, for example near midnight between June 21-22. On that night there is some evidence

that paving operations near the T0 site produced a strong local source of asphalt particles in the coarse
mode with a long tail into the sub-micron regime (Zaveri et al., 2012; Cappa et al., 2016). This short-
duration source of large particles pushed the $R_{eff,f}$ temporarily towards larger values. (The $R_{eff,f}$ changes
from the nephelometer + PSAP at this time were smaller than from the CRD or size distribution
observations. Most likely this reflects the alternating 6-min sampling of the nephelometer and the very
short duration of the event leading to discrepancies in the 1 h average.)
Despite the generally good correspondence between $R_{eff,f,size}$ and the optically derived values, the $R_{eff,f,size}$
values were often (but not always) smaller (Table 2). This is most clearly seen when comparing the
average diurnal profiles of the $R_{eff,f}$ values from the different methods, as shown in Figure 5. All three
$R_{eff,f}$ estimates exhibit similar diurnal behavior at T0, even though the $R_{eff,f}$ from the SDA-FMC method
are larger than $R_{eff,f,size}$. The diurnal variability in the $R_{eff,f}$ is more pronounced at T0 than at T1. The
diurnal trend in the effective radius of the fine mode at T0 from all methods exhibits a minimum at
around mid-day and then an increase to a maximum right near daybreak. Particle number and sizes at
both sites were influenced by frequent regional new particle formation and growth events during CARES
(see Figure S2). The events tended to start in the morning with a sharp increase of 10 - 20 nm particles
followed by growth of these particles to 50 – 100 nm in the afternoon as discussed in Setyan et al.
(2014).  The next day the cycle repeats (on average) with the introduction of the new small particles
which has the effect of decreasing the average particle radius (Setyan et al., 2014). Although observed at
both sites, the new particle formation events had a greater impact on the size distributions at T0,
especially in terms of surface area-weighted size distributions (Figure S3) that determine $R_{eff,f}$. In part,
this is likely because of continued growth of the new particle mode as it transits from T0 to T1. In
addition, for T0 there is a notable mode in the surface-area weighted distribution at ~1 micron that is
most evident in the early morning (Figure S3). This mode has little influence on the $R_{eff,f}$ values
determined from the size distributions, but contributes to the higher optically determined $R_{eff,f}$ values in
the early morning for T0. This mode is much less prevalent at the T1 site, and thus there is better
correspondence between the size-distribution and optical methods.

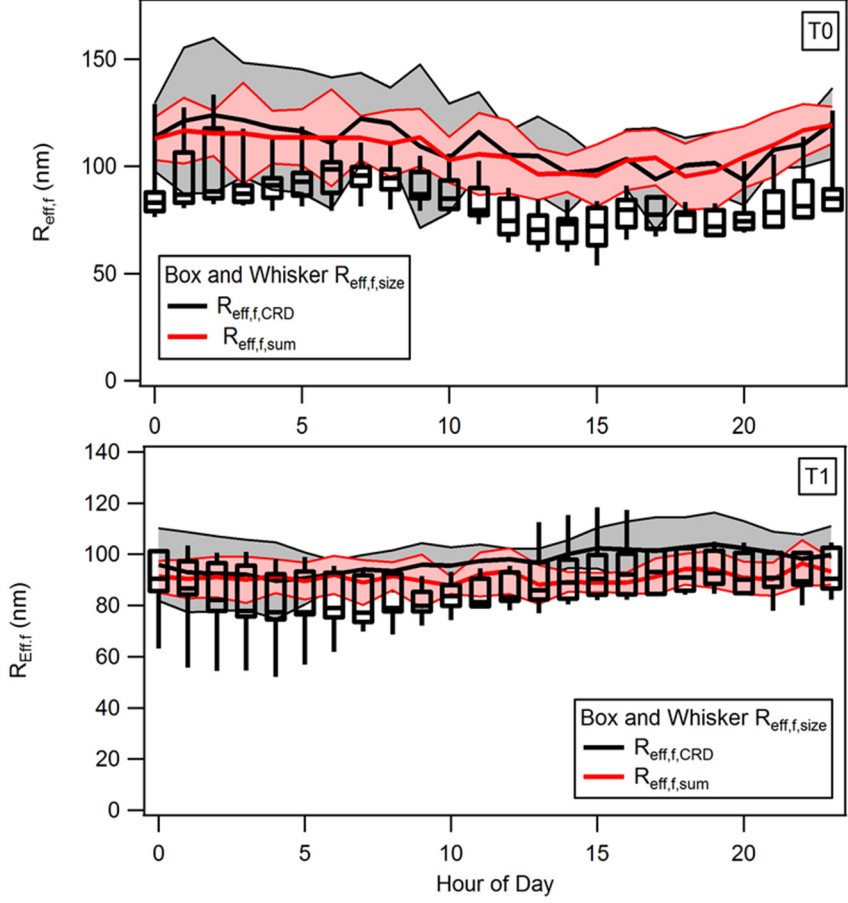

591 **Figure 5** – The diurnal dependence of $R_{eff,f}$ for the period shown in Fig. 4 for the (top) T0 and
(bottom) T1 sites. The box and whisker plot (bottom and top of box are 5% and 95% of data
range, bar is mean, and whiskers extend to full range) shows the results from the direct size
distribution measurement ($R_{eff,f,size}$). The thick lines show the mean diurnal dependence of the
optically derived $R_{eff,f}$, using the CRD (black) and nephelometer + PSAP (red) measurements. The
light colored bands show the $\pm 1\sigma$ standard deviation based on the measurement variability over
the averaging period.
One possible explanation for the differences between the optically and size-derived $R_{eff,f}$, in particular at
T0, may be inaccurate specification of the refractive index. Temporal variations in or an overall offset of
the real refractive index used here from the true value would lead to errors in the optically derived $R_{eff,f}$.
The refractive index is used to convert the derived van de Hulst parameter to $R_{eff,f}$ (Eqn. 3). Given the
form of the relationship, an absolute error in the real RI of 0.1—likely an upper limit—corresponds to an
error in the derived $R_{eff,f}$ of 20%, with larger values of the real RI leading to smaller derived $R_{eff,f}$. The
imaginary component was assumed zero. The effective imaginary RI is likely $\leq 0.01$, given the range of
single scatter albedo values observed (Cappa et al., 2016). Thus, the assumption of zero for the
imaginary RI introduces negligible error. The actual real RI depends on the particle composition since
different chemical components (e.g. sulfate, organics, dust) have different RI values. Here, the RI values
used were determined based only on measurements of the non-refractory PM composition and only an
average value was used (Atkinson et al., 2015). To the extent that refractory components, in particular
dust or sea salt, contributed to the fine mode scattering, their influence on the real RI would not be
accounted for.  However, dust and sea salt contributions are most likely confined primarily to the coarse
mode. Thus, the fine mode real refractive index is unlikely to be strongly affected by their presence and
the real RI can probably be constrained to a fairly narrow range around 1.5. The relative uncertainty of
the $R_{eff,f}$ derived from the SDA-FMC method has been estimated as ranging from 40% to 70%. This range
of values was computed from a quadrature combination of the estimated errors (20-50%)  in the SDA-
FMC retrieval (O'Neill et al., 2003), the CRD measurements (< 5% for the UCD and T0 PSU instrument
and 20% for the T1 PSU instrument) and the refractive index term above (estimated maximum of 20%).
In this context, the agreement shown in Fig. 4 is acceptable and may suggest that the above error
estimates are overly conservative.
**Table 2**: Summary statistics for $R_{eff,f}$ values (nm) and FMF (unitless fraction)

| Site | Method | Maximum | | Minimum | | Mean | | Standard Deviation | |
|------|--------|---------|-----|---------|-----|------|-----|--------------------|-----|
| | | $R_{eff,f}$ (nm) | FMF | $R_{eff,f}$ (nm) | FMF | $R_{eff,f}$ (nm) | FMF | $R_{eff,f}$ (nm) | FMF |
| T0 | SDA-FMC + CRD (PM$_{2.5}$) | 208 | 0.97 | 39 | 0.54 | 110 | 0.79 | 21 | 0.09 |
| T0 | SDA-FMC + Neph. & PSAP (PM$_{10}$) | 153 | 0.85 | 68 | 0.35 | 107 | 0.62 | 14 | 0.12 |
| T0 | Size Distribution Integration | 133 | 0.87 | 54 | 0.34 | 85 | 0.58 | 14 | 0.12 |
| T1 | SDA-FMC + CRD (no size cut) | 176 | 0.89 | 46 | 0.22 | 102 | 0.58 | 18 | 0.16 |
| T1 | SDA-FMC + Neph. & PSAP (PM$_{10}$) | 111 | 0.9 | 76 | 0.24 | 91 | 0.58 | 6 | 0.16 |
| T1 | Size Distribution Integration | 118 | 0.87 | 52 | 0.24 | 88 | 0.61 | 11 | 0.15 |


## Conclusions

This work demonstrates that the use of a non-size-selected, three wavelength CRD measurement in
continuous field monitoring, coupled with the SDA-FMC analysis, can provide information about the
relative contribution of the fine mode to the observed total particle extinction. The retrieved value of
the fine mode fraction of extinction is dependent upon the size range of particles sampled and the
overall nature of the particle size distribution. The relationship between the $FMF_{ext}$ and the $SMF_{ext}$,
determined from near-coincident measurement of extinction by $PM_1$ and $PM_{10}$, provides insights into
the effective $FMF_{ext}$ split size. For one of the sites considered here the split point size is around 1 μm
while for the other it is somewhat larger than 1 μm and perhaps more variable. In many environments,
variability in aerosol properties on short (<10 min) timescales is relatively minimal. In such cases, a single
instrument can be used to sequentially sample $PM_1$ and $PM_{10}$, allowing for *in situ* measurement of both
the $FMF_{ext}$ and $SMF_{ext}$. However, remote sensing measurements characterize only the $FMF_{ext}$, (or at best,
an optically influenced size cut as is done in the AERONET retrievals of Dubovik & King, 2000). Thus,
further consideration of *in situ* measurement results, such as those investigated in this study, can
provide insights into the interpretation of the $FMF_{ext}$ determined from remote sensing in different
environments.
The SDA-FMC approach also allows for determination of the effective fine mode radius. The $R_{eff,f}$
characterizes the surface-area weighted size of the particles within the fine mode distribution. The
similarity of the results in Figure 4 for application of the SDA-FMC to both size-selected and non-size-
selected aerosol as well as the comparison with results derived from the PSD measurements verify that
"whole air" measurements (i.e., no imposed size-selection) can provide reliable fine mode radii at least
for large FMF values.

## **Acknowledgements**

This work was supported by the Atmospheric System Research (ASR) program sponsored by the US
Department of Energy (DOE), Office of Biological and Environmental Research (OBER), including Grant
No. DE-SC0008937. Funding for data collection was provided by the US DOE's Atmospheric Radiation
Measurement (ARM) Program. All data used in this study are available from the ARM data archive at:
http://www.arm.gov/campaigns/aaf2009carbonaerosol. The views expressed in this document are
solely those of the authors and the funding agencies do not endorse any products or commercial
services mentioned in this publication.

## Appendix A – Glossary of Symbols and Acronyms used


å                 Ångström exponent (from wavelength pair)
$\alpha$                 Spectral derivative of optical property
$\alpha'$                Curvature (second derivative of optical property in log-log space)
$\alpha_f$ or $\alpha'_f$        Fine mode version of properties (also coarse mode properties $\alpha_c$)
AOD             Aerosol optical depth
$b_{ext}$, $b_{scat}$, $b_{abs}$    Optical coefficient for extinction, scattering, absorption (inverse length units)
CRD             Cavity ring down
$R_{eff,s}$            Effective radius for fine mode
FMF (aka η)     Fine mode fraction of an optical property, usually extinction
SMF             Sub-micron fraction (particle mode with radius or diameter smaller than 1 μm)
$\rho_{eff,f}$           Effective fine mode van de Hulst parameter (product of refractive index and effective
650                        radius)

SDA             Spectral Deconvolution Algorithm
FMC             Fine Mode Curvature approach
$PM_1$            Particulate matter with diameter (or radius) smaller than 1 μm (also $PM_{2.5}$, $PM_{10}$)
PSAP           Particle soot absorption photometer instrument

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
