# Peer review of "Using Spectral Methods to Obtain Particle Size Information from Optical Data"

_Atmospheric Chemistry and Physics, 2017_

## Referee Comment (RC1) · Anonymous Referee #2 · 17 Nov 2017

Review of 'Using Spectral Methods to Obtain Particle Size Information from Optical Data: Applications to Measurements from CARES 2010', by Atkinson *et al.*, 2017.

**Summary and General Comments**

The work presented by Atkinson *et al*. applies a spectral deconvolution algorithm (SDA) and fine mode curvature (FMC) algorithm for retrieving fine mode fraction (FMF) and effective fine mode radius ($R_{eff,f}$), respectively, from *in situ* optical measurements on aerosol particles. Although these algorithms have been applied previously to remote sensing measurements, the work reported here represents the first application to *in situ* optical measurements, allowing an assessment of the accuracy of the retrievals of FMF and $R_{eff,f}$ through comparisons with other *in situ* measurements that measure FMF and $R_{eff,f}$ in a more direct manner. The *in situ* techniques for measuring aerosol optical properties include cavity ring-down spectroscopy (extinction coefficient), nephelometry (scattering coefficient) and particle soot absorption photometry (PSAP, absorption coefficient), with measurements made at a variety of wavelengths spanning the visible and near infrared, and for aerosol ensembles using a variety of impactor cut sizes (1 µm, 2.5 µm and 10 µm). Moreover, fine mode particle size distributions are measured directly using a scanning mobility particle sizer. The reported assessments of FMF and $R_{eff,f}$ retrieval accuracies are important to those in the remote sensing community and also those seeking to characterise aerosol size distribution properties from *in situ* optical measurements. To this end, the work represents a substantial contribution and is suitable for publication in Atmospheric Chemistry and Physics. I recommend publication after the following comments have been addressed.

**Specific comments**

Line 160: It would be good if the authors could be more specific as to how biomass burning confounds the expectation of an anthopogenic-associated fine mode and a coarse mode associated with natural emissions. In particular, the authors reference Hamill *et al*. 2016, but it would be useful for the authors to be more specific about what this study reported that is relevant to the current argument.

Line 204: Please could the authors explain what is meant by 'polar angle representation of $\alpha_f'$ vs $\alpha_f$'. In particular, it would be useful if this representation could be plotted using some of the extinction data later reported for the reader to visualise. Moreover, the van de Hulst parameter and how it is calculated from optical data using the polar plots referred to should be explained more clearly to provide the reader with greater clarity and tools for understanding the results later in the text. In my view, this is one change that would greatly improve understanding readability and understanding, and simply referring the reader to O'Neil et al. 2005 to get all the necessary theoretical details is not helpful. Perhaps, if such a discussion is too long for the main text, a discussion on the polar representation and example plots could be provided in the supplementary information.

Lines 322 – 326: The best-fit slope is 0.87. I'm surprised that the agreement is not better and be closer to a 1:1 relationship. Is the high noise, associated with the poorer precision in the PSU measurements, responsible for this deviation? Please could the authors describe why the PSU CRDS is less precise and state is clear terms that the data from the PSU instrument is neglected in further analysis here because of this poorer precision. Also, for the aforementioned reasons (poor slope of

0.87 and poor precision in PSU 532-nm CRDS data), I do not agree with the phrase '…the two instruments were measuring the same aerosol with comparable measurement quality…'.

Line 334: What is the basis for an inverse wavelength dependence? A reference showing that inverse wavelength dependence is a reasonable approximation would also be useful here.

Lines 401 – 403: The authors discuss errors in $R_{eff,f}$ (later in the text) that arise in part from 5% errors in cavity ring-down extinction measurements. However, no consideration is given to the uncertainties that arise in FMF or $R_{eff,f}$ from errors in the summation (scattering + absorption) data. Given the very large uncertainties and biases that exist in filter-based measurements of absorption, such as from a PSAP, can the authors comment on the corresponding uncertainties in their FMF and $R_{eff,f}$ retrievals when using the summation method. Have the authors considered the influence of absorption correction schemes for filter-based absorption measurements?

**Technical comments:**

Line 19: To reinforce that the ground based measurements are *in situ* opposed to ground based remote sensing, it would be effect to use the phrase 'Multi-wavelength *in situ…*' in the opening sentence.

Line 24, *'Application to in situ measurements allows for comparison…'*: This is a bit ambiguous. Please can the authors specify what is being applied to the *in situ* measurements (the SDA and FMC algorithms). Also, please specify the quantities being compared when stating '…*for comparison…*'.

Line 78: Brackets are not required. In any case, full stop should be after end bracket rather than before.

Line 79: There is some ambiguity here. Please specify what is meant by 'former' and 'latter'. In part, this ambiguity is magnified by the inclusion of the preceding statement concerning the list of symbols and acronyms.

Line 99: The phrase is brackets is unclear. What does the $\eta$ symbol represent? It doesn't appear in the rest of the text. What is 'ibid'? Also, full stop after the end brackets rather than before.

Line 99 – 101: This sentence is confusing on first read as it suggests that the fine mode spectral derivatives can be used with equation 1 to calculate $R_{eff,f}$. In actual fact, the authors are saying that the spectral derivatives can be used to calculate $R_{eff,f}$ using a fine mode curvature algorithm, although a strict definition of $R_{eff,f}$ in terms of the number size distribution is provided by equation 1. Perhaps, a suitable rewording would be 'The fine mode spectral derivatives can then be used to obtain the effective radius for the fine mode through a fine mode curvature algorithm. Alternatively, the fine mode effective radius can be calculated from direct measurements of size distribution (e.g. from scanning mobility particle sizer) using equation 1 (Hansen and Travis (1974)):'.

Line 108: There is some ambiguity here. Please specify that it is particle size information from SMPS data that is included in the integration.

Line 114: Please specify that the methods are 'Numerical methods'. Also, please amend text to state that these 'numerical methods' are not *for* remote sensing measurement, rather are *applied* to remote sensing data.

Line 138: For readability, the authors might want to move the sentence on lines 143 – 144 to after line 138 '…complementary *in situ* measurements.' to describe the direct measurements of $R_{eff,f}$ that

the authors perform. Also, mention here that optical measurements of impactor-selected portions of the aerosol ensemble were performed to measure FMF directly.

Lines 141 – 143: Brackets not required.

Line 180 – 183: Could the authors make it clearer that $\alpha$ is the spectral derivative for the whole aerosol sample, while $\alpha_f$ is that measured when an impactor is used to remove coarse mode contributions.

Line 181, '…is combined with…': This is ambiguous. How are $\alpha$ and $\alpha_f$ combined? This is unclear to the reader at this early stage in the text. An equation to define FMF in terms of $\alpha$ and $\alpha_f$ would be useful here. Indeed, this equation is equation (2) later in the text. Could the authors move equation 2 to this point and define FMF here.

Line 189: What is meant by 'modality'? Please could the authors clarify the text here.

Line 189: What 'measurements' are the authors referring to? Size distribution measurements, perhaps.

Line 190: Could the authors give these three equations? What are the dependent variables?

Line 188 – 191: This whole sentence is vague, difficult to read and needs clarifying. What is meant by 'approximation level relative to a theoretical Mie representation' and 'limited to second order'?

Line 193: Is the set of three equations referred to here the same as the 'three succinct equations' referred to on line 190? If so, please clarify in the text.

Line 196: Please specify reference wavelength. I believe this is 500 nm, but please specify to remove any doubt.

Line 209: What is 'ibid'?

Line 210: The is ambiguity here. What is meant by 'this' in 'estimate of this purely optical parameter…'. Presumably, 'this' is referring to the van de Hulst parameter, but the authors should be more specific here to remove doubt.

Lines 225 – 226: Ambiguity; it is not clear what is meant by 'polar-coordinate system relationship'. Moreover, the phrase 'near monotonic fit' is also ambiguous; a near monotonic fit of what function?

Line 228: Brackets around reference not needed.

Line 230: Ambiguity; please specify what is being compared in 'The comparisons…'.

Line 246: What is meant by 'expensive'? Computationally expensive, or expensive in monetary terms?

Line 296: Remove brackets around reference.

Line 340: Lower case 'N' in nephelometer.

Line 425: Do the authors mean extinction, instead of scattering? For the cases of aerosols sampled here, it probably does not matter. But, with the authors preferring *extinction* throughout the manuscript, it would be good to be consistent.

Line 538: Full stop (period) required after 'fine mode distribution'.

---

## Referee Comment (RC2) · Anonymous Referee #1 · 20 Nov 2017

General comments The manuscript presented by Atkinson et al. describes the retrieval of particle size related information from multi-wavelength aerosol extinction, scattering and absorption filed measurements using spectral deconvolution method that is typically used in remote sensing applications. The authors aim to compare the retrieved values with values that are calculated directly from size distribution measurements in order to validate the retrieval approach and to discuss its limitations. This work contains substantial contribution to further verification of remote sensing measurements using in-situ instruments. I recommend publication after the following comments have been addressed. Most importantly, as the main goal of this work is to evaluate the spectral deconvolution algorithm by comparison to size distribution

measurements an additional effort should be made by the authors to describe and present the error propagation or uncertainty calculation inherent to each calculation from the uncertainties in each measured parameter. Specific and technical comments 1) Line 232: "…averaged AERONET‐SDA differences of 10% +- 30% for large FMF values > 0.5". It is not clear if the authors mean a difference of -20% to +40% or from 0% to +40%? 2) Line 254: since measurement of aerosols light extinction are by definition only apply to the forward direction it is unclear what the authors mean by truncation errors in CRD? 3) Line 313: data in table 1 regarding the PSU-CRD does not correspond to the text. 4) Line 323: a slope of 0.87 in the correlation between two CRD instruments at the same wavelength is significant. What is the uncertainty on this value? How was this 13% error mitigated in the data analysis? Was any correction applied? And how sure are the authors that the same "error" would apply to the 1064nm or the 405 nm CRD's? The authors are sure that with this 13% difference between the instruments "the two instruments were measuring the same aerosol with comparable measurement quality". I do not agree with this statement. 5) Line 343-350: SMPS scans typically take several minutes. A car passing by or a wind gust will cause significant changes to the aerosols population in time scales of seconds. This can be verified by looking at total aerosols concentration data taken with a CPC with a 1 sec resolution. To overcome mid-scan dramatic changes some dead volume is typically applied to allow for mixing of the aerosols and to smooth rapid changes. What measures were taken to insure that each individual SMPS scan is not interrupted by such events? 6) All figures should have some indication of the uncertainty of the presented values in order for the reader to appreciate the variation in the data within/between data series and temporal variation such as the diurnal cycle. 7) Lines 398-399: the authors claim that the difference between FMFCRD and FMFsum is due to significant contribution by large particle. Wouldn't it be possible to fine some support for this claim in the SMPS data? 8) Lines 419-423: I am afraid I don't understand how the differences between the two sites (and not the difference between CRD and SUM in site T0) "highlights the fact that there is not a precise definition of "fine" and

"coarse" in terms of a specific size cut in the optical method." Additionally it is not clear what do the aouthors mean by the shape of the size distribution. Is it the width and/or amplitude ?  9) In figure 4 errors are needed to establish if the temporal variability is real or within uncertainty. This is important for conclusions presented in lines 461-463 and 477-478.  10) Figure 5: why is the discrepancy mostly clear in the first half of the day then the second half of the day in site T0?

Please also note the supplement to this comment:
https://www.atmos-chem-phys-discuss.net/acp-2017-886/acp-2017-886-RC2-supplement.pdf
* * *

---

## Referee Comment (RC3) · Anonymous Referee #3 · 21 Nov 2017

This manuscript describes a spectral deconvolution and fine mode curvature method that can retrieve particle size and determine relative contribution of the fine mode particles to the total particle extinction from Multi wavelength aerosol extinction, absorption and scattering measurements. Typically this method is used in remote sensing applications but authors extended the application of this method to in-situ measurements to retrieve particle size. The authors used extinction data from cavity ring down measurements, scattering data from nephelometer and absorption data from particle soot absorption photometer measurements. Overall, the manuscript is clearly written, some suggested clarifications are listed below. I understand this is more of a technique based manuscript but little bit more discussion about the science would be useful. I recommend this paper for publication. However, prior to acceptance, the authors should address the following questions/ suggestions and modify the manuscript accordingly.

My main concern here is about the error analysis in the retrieved size and contribution of the fine mode particles to the total particle extinction. What are the errors on the estimates? A range of relative uncertainties are stated towards the end of the manuscript but it is not clear to me if the authors consider propagation of errors from the measurements.

In the abstract the authors should briefly mention the major limitations of the technique instead of just stating "..some limitations are also identified". Some of the limitations are mentioned in the text at different places but I suggest providing a list of all the limitations in details at the end so that it would be easier for readers to follow.

Line 177: please provide detail about the polynomial fit that yields a wavelength invariant version.

Line 220: I think authors should expand the discussion regarding the uncertainty in refractive index. How the estimated size will affect if some of the plumes contain more absorbing particles such as soot? Authors used an average value of real part from previous study. Here authors can propagate the error.

Line 249: Authors mention here about the truncation angel error but it is not clear to me if they incorporated the corrections to the nephelometer data.

Line 253: This part somehow misleading to me "Cavity ring down measurements do not (in principle) need to be calibrated"

Line 254: "have very small truncation errors"- please provide a number here.

Line 310: Authors mentioned about low relative humidly during measurements used here. Was it low also at T1 site? Scattering measurements can be substantially impacted at high RH.

Line 333: "The absorption coefficients were adjusted to the nephelomete wavelengths using an inverse wavelength dependence"- please elaborate.

Error bars should be provided in all the figs.

Line 409: "are very similar in absolute magnitude"-please provide the numbers

Fig.3- FMF-CRD shows higher fine mode fraction during 06/19 to 06/20. Is it because of the no size cut for the CRD measurements?

Please consider to change the scale of the y-axis in Fig. 4. Shorter range would help to visualize the variations.

Fig. 5. Once authors do the error propagation, error bars should be included in the figure. Is it 1 hr average for the retrieved radius? What would be the minimum integration time for the optically derived radius to achieve a reasonable estimate? In other words, if there is a spike in the data for shorter time, can it be captured?

---

## Author Comment (AC1) · 19 Mar 2018

We thank the reviewers for their careful reading of the manuscript and their comments and suggestions. We have addressed each of their queries and believe that the paper is strengthened. Our point-by-point responses to the Reviewers' comments and suggestions follow below.

The reviewer comments are in black and our responses in blue. New text added to the manuscript is *italicized*.

**Reviewer #2**

**Summary and General Comments**

The work presented by Atkinson *et al*. applies a spectral deconvolution algorithm (SDA) and fine mode curvature (FMC) algorithm for retrieving fine mode fraction (FMF) and effective fine mode radius ($Reff,f$), respectively, from *in situ* optical measurements on aerosol particles. Although these algorithms have been applied previously to remote sensing measurements, the work reported here represents the first application to *in situ* optical measurements, allowing an assessment of the accuracy of the retrievals of FMF and $Reff,f$ through comparisons with other *in situ* measurements that measure FMF and $Reff,f$ in a more direct manner. The *in situ* techniques for measuring aerosol optical properties include cavity ring-down spectroscopy (extinction coefficient), nephelometry (scattering coefficient) and particle soot absorption photometry (PSAP, absorption coefficient), with measurements made at a variety of wavelengths spanning the visible and near infrared, and for aerosol ensembles using a variety of impactor cut sizes (1 μm, 2.5 μm and 10 μm). Moreover, fine mode particle size distributions are measured directly using a scanning mobility particle sizer. The reported assessments of FMF and $Reff,f$ retrieval accuracies are important to those in the remote sensing community and also those seeking to characterise aerosol size distribution properties from *in situ* optical measurements. To this end, the work represents a substantial contribution and is suitable for publication in Atmospheric Chemistry and Physics. I recommend publication after the following comments have been addressed.

We thank the reviewer for their comment on the utility of this work towards the remote sensing community.

**Specific comments**
Line 160: It would be good if the authors could be more specific as to how biomass burning confounds the expectation of an anthopogenic-associated fine mode and a coarse mode associated with natural emissions. In particular, the authors reference Hamill *et al*. 2016, but it would be useful for the authors to be more specific about what this study reported that is relevant to the current argument.

We have added the following text to the manuscript to clarify: "*In particular, it can be difficult to distinguish biomass burning particles from particles derived from urban sources, as both primarily fall within the fine mode and are somewhat absorbing.*"

Line 204: Please could the authors explain what is meant by 'polar angle representation of $\alpha f$ vs $\alpha f$. In particular, it would be useful if this representation could be plotted using some of the extinction data later reported for the reader to visualise. Moreover, the van de Hulst parameter and how it is calculated from optical data using the polar plots referred to should be explained more clearly to provide the reader with greater clarity and tools for understanding the results later in the text. In my view, this is one change that would greatly improve understanding readability and understanding, and simply referring the reader to O'Neil et al. 2005 to get all the necessary theoretical details is not helpful. Perhaps, if such a discussion is too long for the main text, a discussion on the polar representation and example plots could be provided in the supplementary information.

We have extensively revised the text near line 204 to attempt to clarify the statement about the polar representation and to provide further details regarding interpretation. We considered adding a figure similar to that shown in O'Neill et al. (2005), shown below. The figure itself is exceptionally complex, and thus we have decided to not include a new figure (either in the main text or supplemental).

[revised manuscript text omitted]

Fig. 1 from O'Neill et al. (2005).

Lines 322 – 326: The best-fit slope is 0.87. I'm surprised that the agreement is not better and be closer to a 1:1 relationship. Is the high noise, associated with the poorer precision in the PSU measurements, responsible for this deviation? Please could the authors describe why the PSU CRDS is less precise and state is clear terms that the data from the PSU instrument is neglected in further analysis here because of this poorer precision. Also, for the aforementioned reasons (poor slope of 0.87 and poor precision in PSU 532-nm CRDS data), I do not agree with the phrase '…the two instruments were measuring the same aerosol with comparable measurement quality…'.

The reviewer raises an important point about comparability between the two instruments. First, we have deleted the phrase mentioned by the reviewer ("…the two instruments…"). Second, more importantly, we note that the original fit was performed using a standard linear regression. However, because there is uncertainty in both the x and y it is more appropriate to use an orthogonal distance regression (ODR) fit. The slope from an ODR fit (performed in Igor Pro using the ODR=2 command) yielded an improved slope of 0.96 and an intercept of -0.2 ± 0.25, i.e. indistinguishable from zero. (We note that this revised slope is consistent with that obtained if a ratio is taken between the measurements two instruments, and then a Gaussian curve is fit to a histogram of the ratios. This indicates the appropriateness of the ODR fit.) This slope is within the measurement uncertainty of the two instruments. The figure and discussion in the text have been updated accordingly.

The difference in precision between the instruments most likely results from differences in instrument design, electronics, alignment and mirror quality. While precision is certainly a concern, for our analysis the accuracy, as assessed by the comparability between the UCD and PSU instruments, is more important. Poorer precision in the PSU measurements will translate to lower precision in the derived FMF and fine mode effective radius. However, the overall trends and the average behavior would be unaffected by the poorer precision, so long as the two instruments agree on average (which they do). We have revised the text as follows, and updated Fig. S1.

*To obtain three-wavelength bext measurements for use in the SDA-FMC analysis, we combined the measurements from the two CRD instruments (the 1064 nm measurements from the PSU instrument were used with the 532 nm and 405 nm UCD data after all had been averaged to one-hour). To assess whether this was a reasonable approach, the 532 nm time series data from the two instruments were overlaid and examined for differences. There is a high degree of temporal correspondence between the measurements from the two instruments, although there was a clear difference in precision, with the UCD CRD having approximately 3 times better precision than the PSU instrument at comparable integration times. This difference in precision results from differences in instrumental design and (likely) mirror quality. A scatterplot (Figure S1) of bext,PSU versus bext,UCD also showed good correlation, with a best fit line from an orthogonal distance regression fit having a slope = 0.96 and an intercept that was statistically indistinguishable from zero. This is within the uncertainties of the instruments. The good agreement at 532 nm between the PSU and UCD instruments suggests that combining the 1064 nm measurements from PSU with the 405 nm and 532 nm measurements from UCD is reasonable. If the very slight low bias in the 532 nm bext from PSU relative to the UCD measurements applies to the 1064 nm measurements then the derived FMF values might be slightly overestimated.*

[Figure]

Line 334: What is the basis for an inverse wavelength dependence? A reference showing that inverse wavelength dependence is a reasonable approximation would also be useful here.

We have modified this to say "The absorption coefficients were *interpolated* to the nephelometer wavelengths *assuming the* inverse wavelength dependence *characteristic of uncoated black carbon as appropriate for this region (Cappa et al., 2016).*"

Lines 401 – 403: The authors discuss errors in $R_{eff,f}$ (later in the text) that arise in part from 5% errors in cavity ring-down extinction measurements. However, no consideration is given to the uncertainties that arise in FMF or $R_{eff,f}$ from errors in the summation (scattering + absorption) data. Given the very large uncertainties and biases that exist in filter-based measurements of absorption, such as from a PSAP, can the authors comment on the corresponding uncertainties in their FMF and $R_{eff,f}$ retrievals when using the summation method. Have the authors considered the influence of absorption correction schemes for filter-based absorption measurements?

As the reviewer notes, absorption measurements from PSAP instruments can be biased, typically high (Cappa et al., 2008;Lack et al., 2008). The campaign average SSA at 532 nm for T0 was 0.87, as measured by the UCD CRD and photoacoustic instrument (Cappa et al., 2016). This is actually very similar to that obtained from the PSAP + Neph (0.89). The literature cited above suggests biases up to perhaps a factor of two are possible, although based on the conditions during CARES lower values would be expected. Assuming a factor of two positive bias in the PSAP absorption, the extinction would change (decrease) by 5%. However, important to the current study, the potential bias in the PSAP is not thought to be especially wavelength dependent. The method used here relies on spectral curvature and not on the absolute extinction. Thus, if all of the extinction measurements were 5% lower then the curvature would be unaffected. Put another way, if there is a systematic, wavelength-independent bias in the measurements then the impact on the derived FMF and $R_{eff,f}$ would be small. If the bias were strongly wavelength dependent, then the resulting FMF and $R_{eff,f}$ would be impacted.

**Technical comments:**
Line 19: To reinforce that the ground based measurements are *in situ* opposed to ground based remote sensing, it would be effect to use the phrase 'Multi-wavelength *in situ*…' in the opening sentence.
Done

Line 24,*'Application to in situ measurements allows for comparison…'*: This is a bit ambiguous. Please can the authors specify what is being applied to the *in situ* measurements (the SDA and FMC algorithms). Also, please specify the quantities being compared when stating '…*for comparison*...'.
Done

Line 78: Brackets are not required. In any case, full stop should be after end bracket rather than before.
Done

Line 79: There is some ambiguity here. Please specify what is meant by 'former' and 'latter'. In part, this ambiguity is magnified by the inclusion of the preceding statement concerning the list of symbols and acronyms.
Done

Line 99: The phrase is brackets is unclear. What does the eta symbol represent? It doesn't appear in the rest of the text. What is 'ibid'? Also, full stop after the end brackets rather than before.

We have removed this parenthetical. (Ibid is used to refer to the previous reference, but we no longer use this in the manuscript.)

Line 99 – 101: This sentence is confusing on first read as it suggests that the fine mode spectral derivatives can be used with equation 1 to calculate *Reff,f*. In actual fact, the authors are saying that the spectral derivatives can be used to calculate *Reff,f* using a fine mode curvature algorithm, although a strict definition of *Reff,f* in terms of the number size distribution is provided by equation 1. Perhaps, a suitable rewording would be 'The fine mode spectral derivatives can then be used to obtain the effective radius for the fine mode through a fine mode curvature algorithm. Alternatively, the fine mode effective radius can be calculated from direct measurements of size distribution (e.g. from scanning mobility particle sizer) using equation 1 (Hansen and Travis (1974)):'.

Thank you for the suggestion. We have adopted the suggested text.

Line 108: There is some ambiguity here. Please specify that it is particle size information from SMPS data that is included in the integration.
Done

Line 114: Please specify that the methods are 'Numerical methods'. Also, please amend text to state that these 'numerical methods' are not *for* remote sensing measurement, rather are *applied* to remote sensing data.
Done

Line 138: For readability, the authors might want to move the sentence on lines 143 – 144 to after line 138 '…complementary *in situ* measurements.' to describe the direct measurements of *Reff,f* that the authors perform. Also, mention here that optical measurements of impactor-selected portions of the aerosol ensemble were performed to measure FMF directly.
Done

Lines 141 – 143: Brackets not required.
Done

Line 180 – 183: Could the authors make it clearer that $\alpha$ is the spectral derivative for the whole aerosol sample, while $\alpha f$ is that measured when an impactor is used to remove coarse mode contributions.

This is a slight misunderstanding. Both come from the optical data – the fine version is a result of the SDA part of the procedure. We have clarified this as follows: "In the SDA-FMC approach, the observed spectral derivative ($\alpha$) is combined with the *SDA-derived* fine mode *spectral derivative* ($\alpha$f) to produce the fine mode fraction of extinction. *T*he fine mode slope and curvature are both used in determining the fine mode effective radius."

Line 181, '…is combined with…': This is ambiguous. How are $\alpha$ and $\alpha f$ combined? This is unclear to the reader at this early stage in the text. An equation to define FMF in terms of $\alpha$ and $\alpha f$ would be useful here. Indeed, this equation is equation (2) later in the text. Could the authors move equation 2 to this point and define FMF here.

We have moved Eqn. 2 up to this point, and clarified the text (see response above).

Line 189: What is meant by 'modality'? Please could the authors clarify the text here.

Referring to the two modes. We have modified to "An assumption of an aerosol *size distribution with two distinct modes* yields…

Line 189: What 'measurements' are the authors referring to? Size distribution measurements, perhaps.

Yes. We have modified the text to make this clearer.

Line 190: Could the authors give these three equations? What are the dependent variables?

We have expanded the discussion slightly here, as discussed above in relation to understanding the curvature.

Line 188 – 191: This whole sentence is vague, difficult to read and needs clarifying. What is meant by 'approximation level relative to a theoretical Mie representation' and 'limited to second order'?

We have clarified this as: "*Specifically, the* equations can be inverted to yield the fine mode spectral derivative, the fine mode curvature ($\alpha$f') and the fine and coarse mode AOD or bext values. *It should be noted that the fitting of a 2nd order polynomial to input AOD or bext spectra is only and approximation relative to a higher order polynomial. The use of a 2nd order polynomial represents a compromise between higher order spectral polynomials being better representations of theoretical Mie spectra and the*

*beneficial damping effects of lower order polynomials in the presence of noisy spectra (O'Neill et al., 2001)."*

Line 193: Is the set of three equations referred to here the same as the 'three succinct equations' referred to on line 190? If so, please clarify in the text.

Please see response to previous comment.

Line 196: Please specify reference wavelength. I believe this is 500 nm, but please specify to remove any doubt.

done

Line 209: What is 'ibid'?

We have removed all references to ibid.

Line 210: The is ambiguity here. What is meant by 'this' in 'estimate of this purely optical parameter…'. Presumably, 'this' is referring to the van de Hulst parameter, but the authors should be more specific here to remove doubt.

This has been clarified.

Lines 225 – 226: Ambiguity; it is not clear what is meant by 'polar-coordinate system relationship'. Moreover, the phrase 'near monotonic fit' is also ambiguous; a near monotonic fit of what function?

We have extensively revised the text near line 204 to attempt to clarify the statement about the polar representation and to provide further details regarding interpretation. We provide above the detailed changes to this section.

Line 228: Brackets around reference not needed.

We have reworded and removed the brackets.

Line 230: Ambiguity; please specify what is being compared in 'The comparisons…'.

We have reworded to clarify. The modified text reads:

> *The sky radiance data are collected* nominally once per hour *while AOD measurements are made* once every 3 minutes. Comparisons *between the FMC method and the D&G inversions* show averaged *FMC versus AERONET* differences of *10% $\pm$ 30% (mean $\pm$ standard deviation of ($\rho_{eff,f,FMC}$ - $\rho_{eff,f,D\&K}$) / $\rho_{eff,f,D\&K}$)* for large FMF values > 0.5, *at least for the limited data set of O'Neill et al. (2005) and confirmed by more recently unpublished AERONET-wide comparisons between the FMC and D&G methods.*

Line 246: What is meant by 'expensive'? Computationally expensive, or expensive in monetary terms?

The latter. As we are discussing physical equipment at this point, we believe the use is sufficiently clear.

Line 296: Remove brackets around reference.

Done.

Line 340: Lower case 'N' in nephelometer.

Done

Line 425: Do the authors mean extinction, instead of scattering? For the cases of aerosols sampled here, it probably does not matter. But, with the authors preferring *extinction* throughout the manuscript, it would be good to be consistent.

We have modified this to read: "Nonetheless, since the major sources of fine and coarse mode particles are likely to be reasonably distinct in many environments, the FMFext,CRD can provide a characterization of the variability in the contributions of such sources to the total *extinction and, in environments where the extinction is dominated by scattering (i.e. when the SSA is large), to the total scattering as well.*"

Line 538: Full stop (period) required after 'fine mode distribution'.

Done

**Reviewer #1**

**General comments**

The manuscript presented by Atkinson et al. describes the retrieval of particle size related information from multi-wavelength aerosol extinction, scattering and absorption filed measurements using spectral deconvolution method that is typically used in remote sensing applications. The authors aim to compare the retrieved values with values that are calculated directly from size distribution measurements in order to validate the retrieval approach and to discuss its limitations. This work contains substantial contribution to further verification of remote sensing measurements using in-situ instruments. I recommend publication after the following comments have been addressed. Most importantly, as the main goal of this work is to evaluate the spectral deconvolution algorithm by comparison to size distribution measurements an additional effort should be made by the authors to describe and present the error propagation or uncertainty calculation inherent to each calculation from the uncertainties in each measured parameter.

We thank the reviewer for the comment about error propagation and uncertainty. We have worked to clarify and add to this aspect of our work.

**Specific and technical comments**

1) Line 232: "…averaged AERONET-SDA differences of 10% +- 30% for large FMF values > 0.5". It is not clear if the authors mean a difference of -20% to +40% or from 0% to +40%?

This has been clarified as:

> "Comparisons between the FMC method and the D&K [Dubovik and King] inversions show averaged FMC versus AERONET differences of 10% $\pm$ 30% (mean $\pm$ standard deviation of $(\rho_{eff,f,FMC} - \rho_{eff,f,D\&K}) / \rho_{eff,f,D\&K}$) for large FMF values > 0.5, at least for the limited data set of O'Neill et al. (2005) and confirmed by more recently unpublished AERONET-wide comparisons between the FMC and D&G methods."

2) Line 254: since measurement of aerosols light extinction are by definition only apply to the forward direction it is unclear what the authors mean by truncation errors in CRD?

We have modified this to:

> "Cavity ring-down measurements *directly quantify total extinction within the cavity, which is contributed from both gases and particles (Smith and Atkinson, 2001; Brown, 2003). To determine extinction by aerosols only, the entering air stream is periodically directed through a filter such that a gas-only reference is determined. Extinction by aerosol particles is determined relative to this gas zero. The aerosol*

*extinction is further corrected to account for the practical aspect that the* complete mirror-to-mirror distance of the optical cavity is typically not filled with aerosols (to keep the mirrors clean) (Langridge et al., 2011).*"*

3) Line 313: data in table 1 regarding the PSU-CRD does not correspond to the text.

We have clarified the capabilities of the PSU-CRD so that the text and table are consistent. The table now indicates that the PSU-CRD measures also at 532 nm. However, it should be noted that for our analysis for the T0 site, the 532 nm data from the UCD CRD-PAS instrument was used, not the PSU-CRD 532 nm data.

4) Line 323: a slope of 0.87 in the correlation between two CRD instruments at the same wavelength is significant. What is the uncertainty on this value? How was this 13% error mitigated in the data analysis? Was any correction applied? And how sure are the authors that the same "error" would apply to the 1064nm or the 405 nm CRD's? The authors are sure that with this 13% difference between the instruments "the two instruments were measuring the same aerosol with comparable measurement quality". I do not agree with this statement.

A similar concern was raised by the previous reviewer. We repeat our response here, and note that we have removed the statement about "measurement quality." Regarding the comparability between 532 nm and 1064 nm, the measurements were made for particles sampled through the same inlet, and thus we expect any differences observed for one channel of this instrument to be similar for the others, given that the main reason for differences between the UCD and PSU CRD instruments is particle losses.

From above: The reviewer raises an important point about comparability between the two instruments. First, we have deleted the phrase mentioned by the reviewer ("…the two instruments…"). Second, more importantly, we note that the original fit was performed using a standard linear regression. However, because there is uncertainty in both the x and y it is more appropriate to use an orthogonal distance regression (ODR) fit. The slope from an ODR fit (performed in Igor Pro using the ODR=2 command) yielded an improved slope of 0.96 and an intercept of -0.2 ± 0.25, i.e. indistinguishable from zero. (We note that this revised slope is consistent with that obtained if a ratio is taken between the measurements two instruments, and then a Gaussian curve is fit to a histogram of the ratios. This indicates the appropriateness of the ODR fit.) This slope is within the measurement uncertainty of the two instruments. The figure and discussion in the text have been updated accordingly.

5) Line 343-350: SMPS scans typically take several minutes. A car passing by or a wind gust will cause significant changes to the aerosols population in time scales of seconds. This can be verified by looking at total aerosols concentration data taken with a CPC with a 1 sec resolution. To overcome mid-scan dramatic changes some dead volume is typically applied to allow for mixing of the aerosols and to smooth rapid changes. What measures were taken to insure that each individual SMPS scan is not interrupted by such events?

First, there is a substantial amount of volume in the sampling masts and the internal plumbing in the trailers, which helps to smooth out fast fluctuations. In looking at the e.g. CPC data (or the extinction observations at their native time resolution of 2 seconds) we find that there are very few periods where plumes, such as that from a car, were sampled. Thus, when the SMPS observations are averaged over an hour, as we have done here, issues related to a single scan will average out. Certainly if we were using each individual SMPS scan, rather than an hour average, plumes would be a larger concern. Further, we note that in Atkinson et al. (2015) we explicitly compared the absolute extinction measurements from to the extinction calculated from the size distribution measurements. Overall, strong linear correlations were observed for the dry extinction with little evidence of outliers that might have resulted from SMPS issues.

6) All figures should have some indication of the uncertainty of the presented values in order for the reader to appreciate the variation in the data within/between data series and temporal variation such as the diurnal cycle.

We have updated the figures to have indications of uncertainty. Further discussion about uncertainties is provided in response to Reviewer #3.

7) Lines 398-399: the authors claim that the difference between FMFCRD and FMFsum is due to significant contribution by large particle. Wouldn't it be possible to fine some support for this claim in the SMPS data?

The SMPS measurements only go up to ~ 800 nm, limiting the ability of the SMPS to provide information on large-particle contributions. However, we note that Kassianov et al. (2012) and Cappa et al. (2016) both discuss at length the large contribution from coarse mode particles to the extinction during CARES. Thus there is very good reason to think that large particles contribute to the difference.

8) Lines 419-423: I am afraid I don't understand how the differences between the two sites (and not the difference between CRD and SUM in site T0) "highlights the fact that there is not a precise definition of "fine" and "coarse" in terms of a specific size cut in the optical method." Additionally it is not clear what do the aouthors mean by the shape of the size distribution. Is it the width and/or amplitude ?

What we mean is that when a property such as "fine mode fraction" is retrieved from remote sensing measurements in different locations or even at different times, the meaning of "fine mode" may change somewhat. The characteristic particle size that distinguishes between those in the "fine" and those in the "coarse" mode is not a constant and will vary based on the particular mix of sources and the nature (e.g. shape, number of actual modes) of the overall size distribution. Also, by "shape" we mean width, position and number of actual modes. We have worked to clarify the discussion as follows:

> "However, *the results demonstrate* that *the optical method does not allow for a* precise definition of "fine" and "coarse" in terms of a specific, *effective* size cut *that*

*distinguishes between the two regimes. While the SMF has an explicitly defined
size cut (PM1), the effective size cut for the FMF can vary.* The effective size cut
is dependent on the *shapes (i.e. widths, positions and number of actual modes) of*
the size distributions in the "fine" and "coarse" size regimes and the extent of
overlap between them, which is dependent on the size range of particles sampled
(e.g. PM2.5 versus PM10). *For remote sensing measurements, the particular size
that distinguishes between the fine and coarse mode therefore likely varies
between locations and seasons."*

9) In figure 4 errors are needed to establish if the temporal variability is real or within
uncertainty. This is important for conclusions presented in lines 461-463 and 477-478.

We have added error bands to Fig. 4. The uncertainties were determined using a Monte
Carlo-type approach in which each input to the calculations was varied randomly and
independently about its mean, and with a weighting determined from the uncertainty in
the input variable.

10) Figure 5: why is the discrepancy mostly clear in the first half of the day then the
second half of the day in site T0?

This likely reflects a shift in the effective size cut associated with the FMF. Below we
show the diurnal profile of the surface-area weighted size distribution. There is clearly a
notable mode right around 1 micron in the early morning/late night periods. When this
contributes substantially, the optically-derived Reff,f is impacted (and shifted towards
larger values) while the size-distribution derived Reff,f is affected to a lesser extent. At
T1 this larger mode is much less evident and thus contributes less to the optically-
derived Reff,f. Overall, the difference has to do with the extent to which the "coarse"
mode penetrates into the "fine" mode. We have added the figure below to the
supplemental material. For these distributions, we have combined the SMPS data with
the APS data. Because of limited data available for the APS at the T0 site (due to an
instrument malfunction) the size distributions are for only a subset of the total period
examined in this manuscript (6/16-6/22). We have added discussion to the main text,
where we already had included discussion related to the nucleation mode that is
observed during the daytime and that also influences the diurnal behavior.

> *"In addition, for T0 there is a notable mode in the surface-area weighted distribution
> at ~1 micron that is most evident in the early morning (Figure S3). This mode has
> little influence on the Reff,f values determined from the size distributions, but
> contributes to the higher optically determined Reff,f values in the early morning for
> T0. This mode is much less prevalent at the T1 site, and thus there is better
> correspondence between the size-distribution and optical methods."*

[Figure]

**Figure S3.** Observed diurnal variation for (left) the T0 site and (right) the T1 site for the surface-area weighted size distribution. Distributions have been normalized to the maximum surface area concentration for each hour of the day. The black box shown for T0 highlights the presence of a mode near 1 micron.

**Reviewer #3**

This manuscript describes a spectral deconvolution and fine mode curvature method that can retrieve particle size and determine relative contribution of the fine mode particles to the total particle extinction from Multi wavelength aerosol extinction, absorption and scattering measurements. Typically this method is used in remote sensing applications but authors extended the application of this method to in-situ measurements to retrieve particle size. The authors used extinction data from cavity ring down measurements, scattering data from nephelometer and absorption data from particle soot absorption photometer measurements. Overall, the manuscript is clearly written, some suggested clarifications are listed below. I understand this is more of a technique based manuscript but little bit more discussion about the science would be useful. I recommend this paper for publication. However, prior to acceptance, the authors should address the following questions/ suggestions and modify the manuscript accordingly.

My main concern here is about the error analysis in the retrieved size and contribution of the fine mode particles to the total particle extinction. What are the errors on the estimates? A range of relative uncertainties are stated towards the end of the manuscript but it is not clear to me if the authors consider propagation of errors from the measurements.

*We thank the reviewer for pushing us to consider our uncertainties to a greater extent. In response, we have added the following text as a new section and updated the figures.*

> *The uncertainty in the SMF has been estimated from standard error propagation of the uncertainties in the $PM_1$ and $PM_{10}$ extinction measurements. The assumed uncertainties in $b_{ext,PM1}$ and $b_{ext,PM10}$ are $\pm 1$ Mm$^{-1}$. This uncertainty estimate accounts only for random errors, not systematic errors.*

> *Uncertainties in the FMF have been estimated based on the uncertainties in the inputs to the SDA-FMC procedure, namely the $b_{ext}$ values. The assumed uncertainties in the input $b_{ext}$ were instrument specific: <1 Mm$^{-1}$ for the UCD CRD, 1 Mm$^{-1}$ for the nephelometer plus PSAP and PSU CRD at T0, and 3 Mm$^{-1}$ for the PSU CRD at T1. The input uncertainties are propagated through the various mathematical relationships using standard methods. The FMF error estimate includes some of the factors that contribute systematic uncertainty in the method. As noted in the Theoretical Approach section, FMF values from the SDA-FMC procedure have been shown to agree well with those determined from the more comprehensive inversion method of Dubovik and King (2000).*

> *Uncertainties in the derived $R_{eff,f}$ are also estimated from the uncertainties in the input values. The size-distribution derived $R_{eff,f}$ values depend on the SMPS measurements. The SMPS instruments were calibrated (using 200 nm polystyrene latex spheres) prior to the campaign and*

*a drier was used to keep the aerosol RH < 30% throughout the entire campaign. Periodic checks throughout the campaign indicate consistent sizing performance to within 5%. The size distribution data used here were corrected for DMA transfer function, the bipolar charge distribution, the CPC efficiency and internal diffusion losses. Under these conditions the estimated uncertainties for $D_p$ are around 10% for the size range between 20 and 200 nm (Wiedensohler et al., 2012). Although larger uncertainties could exist for smaller and larger particle sizes, the derived $R_{eff,f}$ values fell primarily in this range. The estimated SMPS uncertainty (Wiedensohler et al., 2012) was estimated based on intercomparisons between different SMPS instruments and thus probably represents both determinate and indeterminate errors. The relative uncertainty in the $R_{eff,f}$ from the size distribution measurement is thus estimated to be 10%. This estimate mainly reflects uncertainties in the absolute size, since there is expected to be significant cancellation in the errors produced by the particle counter (the same data are used in the numerator and denominator of Eq. 1).*

*Estimating the uncertainty in the $R_{eff,f}$ from the SDA-FMC is more challenging because the uncertainties cannot be simply propagated through the equations. Therefore, an approach was taken wherein a large number of $R_{eff,f}$ values were calculated from input $b_{ext}$ that were independently, randomly varied within one standard deviation of the measured value, assuming a normal distribution of errors. Potential uncertainty or variability in the real refractive index was accounted for based on the compositional variation (Atkinson et al., 2015) and assuming volume mixing applies. The standard deviation (1s) was 0.015. This is likely a lower estimate of the uncertainty in the RI, as it does not account for absolute uncertainty in the estimate. The standard deviation of the derived $R_{eff,f}$ is taken as the uncertainty. This Monte Carlo-style approach does not incorporate systematic error sources. The relative uncertainty in the derived $R_{eff,f}$ is found to range from a few percent up to 40%, depending on the particular instrument suite considered and measurement period. In general, the uncertainties were larger for the PSAP and nephelometer, presumably because the wavelengths used are more closely spaced.*

In the abstract the authors should briefly mention the major limitations of the technique instead of just stating "..some limitations are also identified". Some of the limitations are mentioned in the text at different places but I suggest providing a list of all the limitations in details at the end so that it would be easier for readers to follow.

We have updated the abstract as follows:

"*Overall*, the retrieved fine mode fraction and effective radius compare well with other in situ measurements, including size distribution measurements and scattering and absorption measurements made separately for PM1 and PM10, *although there were* some *periods during which the different methods yielded different results. One key reason identified as contributing to differences between methods is the imprecise definition of "fine" and "coarse" mode from the optical methods, relative to instruments that use a physically defined cut-point.*"

Line 177: please provide detail about the polynomial fit that yields a wavelength invariant version.

We have made substantial revisions to this section, as documented in our response to Reviewer #2 above.

Line 220: I think authors should expand the discussion regarding the uncertainty in refractive index. How the estimated size will affect if some of the plumes contain more absorbing particles such as soot? Authors used an average value of real part from previous study. Here authors can propagate the error.

We are using 1h averages, so very short plumes with highly absorbing material will have little influence on the results. If we look at a histogram of SSA values (see below), we see that there is a reasonably narrow distribution with the vast majority of points between 0.8 and 0.95. Using Mie theory as a guide, we find that the imaginary part of the refractive index need only vary from ~0.004 to 0.02 to produce SSA values in this range. Such variations have a very small impact on the extinction wavelength dependence; it is much more dependent on the real component. That the results are more sensitive to variations in the real part was stated in the manuscript previously: "For example if the composition shifted from pure sulfate aerosol (m = 1.53 + 0i) to a brown carbon organic (m = 1.4 + 0.03i) this would introduce a 33% shift in the derived radius with no change in actual size; the majority of this shift in the derived radius results from the change in the real component of the refractive index."

[Figure]

Line 249: Authors mention here about the truncation angel error but it is not clear to me if they incorporated the corrections to the nephelometer data.

We now state: "The scattering coefficients were corrected for truncation error (Anderson and Ogren, 1998) and the absorption coefficients for filter effects (Ogren, 2010)."

Line 253: This part somehow misleading to me "Cavity ring down measurements do not (in principle) need to be calibrated"

*We have modified this to: "Cavity ring-down measurements directly quantify total extinction within the cavity, which is contributed from both gases and particles (Smith and Atkinson, 2001; Brown, 2003). To determine extinction by aerosols only, the entering air stream is periodically directed through a filter such that a gas-only reference is determined. Extinction by aerosol particles is determined relative to this gas zero. The aerosol extinction is further corrected to account for the practical aspect that the* complete mirror-to-mirror distance of the optical cavity is typically not filled with aerosols (to keep the mirrors clean) (Langridge et al., 2011*)."*

Line 254: "have very small truncation errors"- please provide a number here.

This has been revised. See response to previous query.

Line 310: Authors mentioned about low relative humidly during measurements used here. Was it low also at T1 site? Scattering measurements can be substantially impacted at high RH.

Yes, the RH was low at both sites throughout the campaign, as shown in Zaveri et al. (2012). Something to this effect was mentioned on Line 359: "As with the T0 PSU instrument, the total aerosol system attempts to measure particle extinction at nearly ambient conditions, resulting in low RH (25 – 40 %) throughout most of the campaign, as measured by an integrated RH/T sensor (Vaisala HMP70)."

Line 333: "The absorption coefficients were adjusted to the nephelomete wavelengths using an inverse wavelength dependence"- please elaborate.

We have clarified that the absorption coefficients were interpolated, rather than adjusted.

Error bars should be provided in all the figs.

We have updated the figures to include uncertainty estimates.

Line 409: "are very similar in absolute magnitude"-please provide the numbers

Values are now provided in Table 2.

Fig.3- FMF-CRD shows higher fine mode fraction during 06/19 to 06/20. Is it because of the no size cut for the CRD measurements?

During this period the absolute extinction was particularly low, making it challenging to assess. The reviewer's suggestion is certainly possible. However, we note that with the uncertainties added to the figure it is now apparent that the measurements are the same within the estimated uncertainties.

Please consider to change the scale of the y-axis in Fig. 4. Shorter range would help to visualize the variations.

While we understand the reviewer's suggestion to change the range, we have chosen to maintain the y-axes scales they were, namely varying over the same range for both panels. We have done this to facilitate comparison between the sites.

Fig. 5. Once authors do the error propagation, error bars should be included in the figure.

Error bands have been added (see below).

[Figure]

**Figure 5** – The diurnal dependence of $R_{eff,f}$ for the period shown in Fig. 4 for the (top) T0 and (bottom) T1 sites. The box and whisker plot (bottom and top of box are 5% and 95% of data range, bar is mean, and whiskers extend to full range) shows the results from the direct size distribution measurement ($R_{eff,f,size}$). The thick lines show the mean diurnal dependence of the optically derived $R_{eff,f}$, using the CRD (black) and nephelometer + PSAP (red) measurements. The

light colored bands show the ±1σ standard deviation based on the measurement variability over the averaging period.

Is it 1-hr average for the retrieved radius? What would be the minimum integration time for the optically derived radius to achieve a reasonable estimate? In other words, if there is a spike in the data for shorter time, can it be captured?

Yes, it is possible to retrieve estimates of the Reff,f at higher time resolution and capture spikes. The results at one hour averaging were selected after verifying that the results were not qualitatively different from those with shorter time-scales. We have chosen to focus on the longer term averages, given that remote sensing observations are often used to develop longer-term climatologies for regions and occur in remote regions where short-term spikes are less common. However, in principle shorter time scales can be accessed.